# ALBA proteins facilitate cytoplasmic YTHDF-mediated reading of m6A in *Arabidopsis*

Marlene Reichel [1,2,6], Mathias Due Tankmar[1,6], Sarah Rennie [3,6✉], Laura Arribas-Hernández [1,5], Martin Lewinski [2], Tino Köster [2], Naiqi Wang [4], Anthony A Millar [4✉], Dorothee Staiger [2✉] & Peter Brodersen [1✉]

## Abstract

N6-methyladenosine (m6A) exerts many of its regulatory effects on eukaryotic mRNAs by recruiting cytoplasmic YT521-B homology-domain family (YTHDF) proteins. Here, we show that in *Arabidopsis thaliana*, the interaction between m6A and the major YTHDF protein ECT2 also involves the mRNA-binding ALBA protein family. ALBA and YTHDF proteins physically associate via a deeply conserved short linear motif in the intrinsically disordered region of YTHDF proteins and their mRNA target sets overlap, with ALBA4 binding sites being juxtaposed to m6A sites. These binding sites correspond to pyrimidine-rich elements previously found to be important for m6A binding to ECT2. Accordingly, both the biological functions of ECT2, and its binding to m6A targets in vivo, require ALBA association. Our results introduce the YTHDF-ALBA complex as the functional cytoplasmic m6A-reader in *Arabidopsis*, and define a molecular foundation for the concept of facilitated m6A reading, which increases the potential for combinatorial control of biological m6A effects.

**Keywords** N6-Methyladenosine (m6A); YTHDF Proteins; ECT2; ALBA Proteins; Intrinsically Disordered Regions (IDR)
**Subject Categories** Plant Biology; RNA Biology

## Introduction

*N6*-methyladenosine (m6A) occurs widely in eukaryotic mRNAs. It is introduced into pre-mRNA during transcription in adenosines in DR**A**CH/GG**A**U (D = A/G/U, R = A/G, H = A/C/U) motifs by a deeply conserved RNA polymerase II-coupled methyltransferase complex (Balacco and Soller, 2019). m6A is required to complete embryogenesis in vertebrates and plants (Geula et al, 2015; Zhong et al, 2008). It is also important for yeast sporulation (Clancy et al, 2002) and for sex determination and neuronal development in

insects (Haussmann et al, 2016; Lence et al, 2016). Many developmental functions of m6A rely on cytoplasmic RNA-binding proteins (RBPs) specialized for m6A recognition, or "reading", via a YTH domain (Arribas-Hernández et al, 2018; Ivanova et al, 2017; Kontur et al, 2020; Lasman et al, 2020). These YTH domain family (YTHDF) m6A readers contain the YTH domain at the C-terminus, preceded by a long intrinsically disordered region (IDR) (Patil et al, 2018).

Higher plants encode an expanded family of YTHDF proteins with, for instance, 11 members in *Arabidopsis thaliana* (Arabidopsis) (Fray and Simpson, 2015; Scutenaire et al, 2018). They are called EVOLUTIONARILY CONSERVED C-TERMINAL REGION 1-11 (ECT1-11) with reference to the deeply conserved YTH domain at the C-terminus (Ok et al, 2005), following an IDR of more variable length and sequence. ECT2 and ECT3 are crucial for post-embryonic development (Arribas-Hernández et al, 2018; Arribas-Hernández et al, 2020), as they stimulate cell division in primordial cells (Arribas-Hernández et al, 2020). Thus, double knockout of *ECT2* and *ECT3* causes the slow formation and aberrant morphology of leaves, roots, stems, flowers, and fruits, and these phenotypes are generally exacerbated by additional knockout of *ECT4* (Arribas-Hernández et al, 2020). The developmental role of the m6A-ECT module is conserved in plants, because the knockout of tomato and rice *ECT* genes also causes delayed development (Ma et al, 2022; Yin et al, 2022).

Three features of the molecular functions of ECT proteins that promote growth during organogenesis have been defined. First, they are deeply conserved, because the sole YTHDF protein encoded by the liverwort *Marchantia polymorpha* that diverged from higher plants ~450 million years ago (Magallón et al, 2013; Su et al, 2021) can functionally replace Arabidopsis ECT2 when expressed in primordial cells in *ect2 ect3 ect4* mutants (Flores-Téllez et al, 2023). Second, ECTs interact with the major cytoplasmic poly(A)-binding proteins PAB2/4/8 (Song et al, 2023; Tankmar et al, 2023). This interaction is mediated by a conserved tyrosine-rich motif in the IDR of ECT2 and is required for developmental functions of ECT2 (Tankmar et al, 2023). Third, most Arabidopsis *ECT* paralogues across phylogenetic subclades retain the ability to complement *ect2 ect3 ect4* mutants upon ectopic expression in primordial cells (Flores-Téllez et al, 2023).

[1]University of Copenhagen, Copenhagen Plant Science Center, Department of Biology, Copenhagen N, Denmark. [2]Department of RNA Biology and Molecular Physiology, Faculty of Biology, Bielefeld University, D-33615 Bielefeld, Germany. [3]Department of Biology, Copenhagen University, Copenhagen N, Denmark. [4]Division of Plant Science, Research School of Biology, The Australian National University, Canberra, ACT 2601, Australia. [5]Present address: Consejo Superior de Investigaciones Científicas, Instituto de Hortofruticultura Subtropical y Mediterránea 'La Mayora', Málaga, Spain. [6]These authors contributed equally: Marlene Reichel, Mathias Due Tankmar, Sarah Rennie.
✉E-mail: sarah.rennie@bio.ku.dk; tony.millar@anu.edu.au; dorothee.staiger@uni-bielefeld.de; pbrodersen@bio.ku.dk

For the three Arabidopsis ECT proteins unable to perform this basal function, the divergence can at least in part be ascribed to differences in their N-terminal IDRs (Flores-Téllez et al, 2023), including the loss of the PAB2/4/8-interacting motif (Tankmar et al, 2023). Thus, the molecular properties of the IDRs of ECT proteins are central to understand their biological functions.

At least three distinct molecular properties of IDRs in RBPs are expected to contribute to their functions. First, IDRs often mediate self-assembly such that above a critical concentration, they condense into a phase distinct from the aqueous solution (Wiedner and Giudice, 2021). This is also the case for plant ECT proteins (Arribas-Hernández et al, 2018; Scutenaire et al, 2018), and negative feedback regulation of important stress-related m⁶A-containing mRNAs may indeed rely on ECT-mediated phase separation (Lee et al, 2023; Wu et al, 2024). Second, the IDR may influence RNA-binding activity, either by stabilization of the RNA-bound conformation of the globular RNA-binding domain (Stowell et al, 2018), or through direct RNA-binding activity, as is well-described in the case of Arg-Gly-Gly (RGG) repeats in IDRs (Chong et al, 2018). The non-RGG-containing IDR of ECT2 also has direct RNA-binding activity. Recently reported in vitro binding assays establish that ECT2 only has appreciable m⁶A-binding activity when its YTH domain is combined with elements of the N-terminal IDR (Seigneurin-Berny et al, 2024), consistent with the observation that deletion of the IDR from ECT2 strongly reduces RNA-binding capacity in vivo (Tankmar et al, 2023), and with observations of ECT2 crosslinks to target mRNAs specific to the IDR in crosslinking-immunoprecipitation-sequencing (CLIP-seq) data (Arribas-Hernández et al, 2021a). Third, short linear motifs (SLiMs) may be used to mediate direct binding to other proteins (Holehouse and Kragelund, 2024), including other RBPs and regulators of the rate of translation and mRNA decay, as in the example of the ECT2-PAB2/4/8 interaction (Tankmar et al, 2023).

The ALBA (<u>a</u>cetylation <u>l</u>owers <u>b</u>inding <u>a</u>ffinity) family of proteins was found in mRNA interactome capture screens to be a prominent group of mRNA-associated RBPs in Arabidopsis (Marondedze et al, 2016; Reichel et al, 2016). The ALBA superfamily of proteins contains an archaeal and two eukaryotic families. Proteins in the Sac10b archaeal family (Aravind et al, 2003) exhibit acetylation-sensitive DNA-binding activity and have histone-like properties (Bell et al, 2002; Forterre et al, 1999; Wardleworth et al, 2002; Xue et al, 2000), but may also have RNA chaperone functions (Zhang et al, 2020). The two eukaryotic families group around two distinct subunits of RNaseP/MRP complexes, Rpp20 or Rpp25 (Aravind et al, 2003). Plants encode ALBA proteins belonging to both eukaryotic families. The Rpp20-related forms are short and contain only the ~95 amino acid globular ALBA domain, while the Rpp25-related forms are long and contain ~200-300 amino acid C-terminal extensions, often IDRs with many RGG repeats (Goyal et al, 2016). The sequence similarity within the eukaryotic families is limited, and in most cases, it is not clear whether the ALBA proteins are mRNA-binding or have other RNA-related functions. mRNA-binding ALBA proteins have been studied in the parasitic protist *Trypanosoma brucei*, where short and long forms are required for translational regulation of many mRNAs during the transition between mammalian and insect hosts, in particular for growth after commitment to differentiation into the insect-specific form (Bevkal et al, 2021; Mani et al, 2011).

A requirement of ALBA proteins for growth is recurrent in several plant species (Honkanen et al, 2016; Magwanga et al, 2019), first observed in the liverwort *M. polymorpha* where the sole long

RGG-repeat-containing ALBA protein is necessary for the development of root-like structures called rhizoids (Honkanen et al, 2016). Arabidopsis encodes three short ALBA proteins in the Rpp20 group, ALBA1-3, and three long ALBA proteins in the Rpp25 group, ALBA4-6 (Goyal et al, 2016). Single knockouts of *ALBA1* and *ALBA2* cause defective root hair development, but no overall growth defects (Honkanen et al, 2016). In contrast, combined knockout of *ALBA4-6* leads to slow seedling development, including defective root growth (Tong et al, 2022). A similar defect in root growth was also observed in cotton upon RNAi-mediated knockdown of *ALBA* genes (Magwanga et al, 2019), further supporting the idea that ALBA proteins stimulate tissue growth in plants. Nonetheless, the molecular basis for their growth-promoting function has not been defined.

In this study, we show that ALBAs and ECT2 associate via a deeply conserved SLiM in the IDR of ECT2 to form an efficient m⁶A reader complex in Arabidopsis. The mRNA target sets of ALBA proteins overlap significantly with those of m⁶A-ECT2/3, and ALBA4 binding sites in 3′-UTRs are juxtaposed to m⁶A sites. Finally, ALBA proteins facilitate the association of ECT2 with m⁶A-modified transcripts and are necessary for the biological functions of m⁶A-ECT2/3. Thus, our results uncover a mechanism for facilitated m⁶A reading by YTHDF-interacting RBPs with binding sites in close proximity to m⁶A.

## Results

### The N8 IDR-element of ECT2 is required for normal growth of leaf primordia

We previously showed that a 37-amino acid residue region in the N-terminal IDR of ECT2, N8, is required for full activity in promoting the growth of leaf primordia (Tankmar et al, 2023). Since deletion of the N8-encoding region from an *ECT2-mCherry* gDNA transgene caused a decrease, not abolishment, of the complementation frequency of the *ect2-1 ect3-1 ect4-2* (henceforth, *te234*) triple knockout mutant (Arribas-Hernández et al, 2018; Tankmar et al, 2023), we first sought to corroborate the importance of N8 by independent means. To this end, we used CRISPR-Cas9 in the *ect3-1 ect4-2* genetic background to generate a chromosomal in-frame *ECT2* deletion matching almost exactly ΔN8 (*ect2-5*, Fig. 1A; Appendix Fig. S1). The resulting *ect2-5 ect3-1 ect4-2* mutant exhibited slow emergence of the first true leaves, albeit less pronounced than *te234* (Fig. 1B,C). These results verify that deletion of N8 causes partial loss of ECT2 function. We also confirmed that the ECT2-5 protein accumulated to levels similar to the wild-type protein (Fig. 1D), excluding the possibility that the partial loss of ECT2 function in *ect2-5* mutants is due to decreased dosage.

### N8 is necessary for full RNA association of ECT2

We next conducted in vivo UV crosslinking and immunoprecipitation (CLIP) experiments to test whether RNA association was affected by the deletion of N8. We quantified crosslinked RNA immunoprecipitated with ECT2^WT-mCherry or ECT2^ΔN8-mCherry by polynucleotide kinase (PNK)-mediated radiolabeling, using the previously described assay conditions that allow assignment of the radiolabeled species as ECT2-mCherry-RNA complexes with different sizes resulting from cleavage of the IDR in the lysis

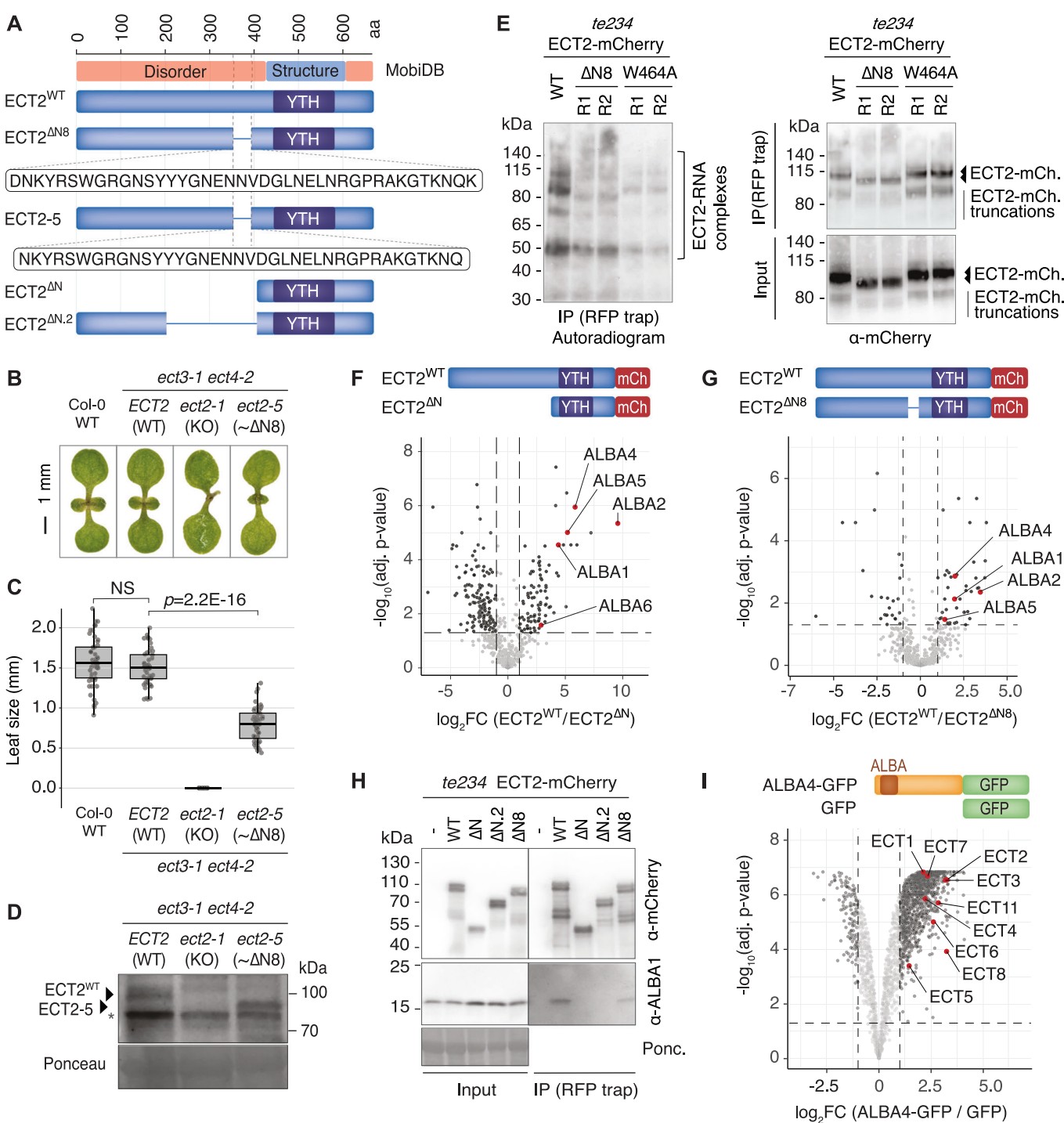

buffer (Arribas-Hernández et al, 2021a). These experiments revealed a reproducible reduction in RNA association of ECT2^ΔN8-mCherry compared to ECT2-mCherry, albeit less pronounced than the reduction obtained with the m⁶A-binding deficient ECT2^W464A-mCherry mutant (Arribas-Hernández et al, 2021a) (Fig. 1E). Thus, N8 is involved in RNA association, either directly or through interaction with other RBPs whose presence may enhance the affinity of ECT2 for m⁶A-containing mRNAs.

## N8 is necessary for interaction with ALBA proteins

To test whether N8 is required for the association of ECT2 with other RBPs, we used comparative immunoprecipitation-mass spectrometry (IP-MS) with stable transgenic lines expressing comparable amounts of either ECT2^WT-mCherry or ECT2^ΔN8-mCherry in the *te234* background ((Tankmar et al, 2023), Appendix Fig. S2A). We also included three lines of ECT2^ΔN-mCherry lacking the entire N-terminal IDR ((Tankmar et al, 2023), Fig. 1A) as an additional negative control.

**Figure 1.    The N8 IDR-element of ECT2 is required for growth promotion, RNA association, and interaction with ALBA proteins.**

(A) Schematic representation of wild type and mutant ECT2 proteins. The MobiDB (Di Domenico et al, 2012) track (top) displays regions predicted to be structured or disordered. (B) Images of representative seedlings of the indicated genotypes taken at 7 days after germination (DAG). (C) Quantification of first true leaf size in seedlings of the indicated genotypes 7 DAG. 50 seedlings were measured for each genotype ($n = 50$). The boxes show the interquartile range (25th–75th percentile), with the central line marking the median. Whiskers extend 1.5 times the interquartile range. Statistical differences between the indicated genotypes were calculated using a Student's *t*-test. NS not significant. (D) Protein blots of total lysates prepared from 12-day-old seedlings of the indicated genotypes, were probed with ECT2-specific antisera (Arribas-Hernández et al, 2018). Arrows indicate the positions of the ECT2$^{WT}$ protein and the ECT2-5 protein containing the N8-like deletion. The asterisk indicates an unspecific band. Ponceau staining serves as the loading control. (E) Results of an in vivo UV crosslinking-ECT2-mCherry-immunoprecipitation experiment, followed by PNK-labeling of precipitated RNA with γ-$^{32}$P-ATP. Left panel, autoradiogram of $^{32}$P-radiolabelled RNA-protein complexes purified from plants expressing ECT2$^{WT}$-mCherry, ECT2$^{\Delta N8}$-mCherry or the aromatic cage mutant ECT2$^{W464A}$-mCherry. Molecular weight marker positions and the location of the verified ECT2-mCherry-RNA complexes (Arribas-Hernández et al, 2021a) are indicated. The presence of several bands of unequal intensity is due to partial proteolysis of the ECT2 IDR during immunoprecipitation and differential labeling efficiency of the different RNPs (Arribas-Hernández et al, 2021a). Right panels, mCherry immunoblots of the immunoprecipitated (top) and total fractions (input, bottom). Samples were pools of three independent lines for each genotype. (F, G) Volcano plots showing the differential abundance of proteins co-purified with ECT2-mCherry variants (RFP-trap) measured by mass spectrometry of immunopurified fractions (IP-MS). All ECT2-mCherry variants were expressed in the *te234* mutant background. Diagrams above each plot indicate the proteins compared. Statistical significance was determined using empirical Bayes statistics with Benjamin–Hochberg adjusted *P* values. Experiments were done in biological triplicates. The data underlying the plot in (F) have previously been published (Tankmar et al, 2023). (H) Co-immunoprecipitation assay using mCherry immunoprecipitation from 10-day-old seedlings expressing the indicated ECT2-mCherry variants (see (A)), followed by immunoblot analysis with mCherry- and ALBA1-specific antibodies. Seedlings from three independent transgenic lines were pooled in this experiment. (I) Volcano plots showing the differential abundance of proteins co-purified with ALBA4-GFP as determined by IP-MS from total lysates prepared from 7-day-old seedlings. Statistical significance was calculated using empirical Bayes statistics with Benjamini–Hochberg adjusted *p* values. The experiment was done in biological triplicates. Source data are available online for this figure.

All immunopurifications were done in the presence of RNaseA to recover RNA-independent interactors. These experiments revealed that the family of ALBA proteins, in particular ALBA1/2/4/5, were prominent interactors of ECT2 (Fig. 1F; Appendix Fig. S2C), and that the interaction was strongly dependent on N8 (Fig. 1G; Dataset EV1). Because of these qualities of ECT-ALBA co-purifications, and because the mRNA-binding capacity of ALBA proteins may be of interest in connection with the requirement of N8 for ECT2 binding to RNA in vivo, we focused further experiments on the ALBA protein family.

We used three different approaches to verify the ALBA-ECT interaction and its dependence on N8. First, we raised an antibody specific for ALBA1 (Appendix Fig. S2B) and used it to confirm that ALBA1 enrichment is reduced, but not abolished, upon deletion of N8 (Fig. 1H). We also included two larger IDR deletion mutants in this experiment, ECT2$^{\Delta N}$-mCherry and ECT2$^{\Delta N.2}$-mCherry lacking the ~200 amino acid residues proximal to the YTH domain ((Tankmar et al, 2023), Fig. 1A,H). ALBA1 levels were not detectable in immunopurified fractions of these two mutants (Fig. 1H), perhaps suggesting that additional determinants of ALBA interaction are located in the IDR outside of the N8 region. Second, an inspection of IP-MS data with HA-ECT2 and with tagged versions of the two YTHDF paralogs ECT3 (ECT3-Venus) and ECT1 (ECT1-TFP) (Tankmar et al, 2023), both of which have m⁶A-binding capacity (Arribas-Hernández et al, 2018; Arribas-Hernández et al, 2021a; Arribas-Hernández et al, 2021b; Flores-Téllez et al, 2023; Lee et al, 2023), revealed enrichment of ALBA proteins over the negative controls (Appendix Fig. S2C). Third, comparative IP-MS analysis carried out with ALBA4-GFP, and free GFP revealed a clear enrichment of several ECT proteins, including ECT1-8 and ECT11, in the ALBA4-GFP purified fractions (Fig. 1I; Appendix Fig. S2D; Dataset EV1). These results indicate that ALBA and ECT proteins physically associate in vivo and that the ECT2-ALBA association involves the N8 region of the ECT2 IDR. We also take particular note of the combination of two properties. First, deletion of N8 causes reduced RNA binding of ECT2 in vivo. Second, ECT interactors of ALBAs include ECT1 and ECT11 which have m⁶A-binding capacity but not the function of ECT2 required for leaf formation (Flores-Téllez et al, 2023). Hence, our results

suggest that the ALBA-ECT interaction mediates a molecular property common to all ECT proteins, perhaps m⁶A-binding.

## AlphaFold3 modeling highlights a conserved SLiM in N8 as key for the interaction of ECT2 with ALBA domains and RNA

Because many proteins, in addition to ALBA1/2/4/5, lose enrichment in immunopurified ECT2 fractions upon deletion of N8 (Fig. 1G), we sought to further narrow the region in the IDR of ECT2 required for ALBA interaction. We noticed that a SLiM within N8 is conserved in all YTHDF protein clades of flowering plants as well as in YTHDFs from early-diverging clades of land plants such as bryophytes, including *M. polymorpha* YTHDF (Fig. 2A; Appendix Fig. S3). Since the N8 region is required for full association of ECT2 with both mRNA and ALBA proteins in vivo, we hypothesized that the N8 element might mediate an interaction between the three molecules, perhaps via the conserved SLiM. Thus, we used AlphaFold3 (Abramson et al, 2024) to query whether a complex composed of an ALBA-domain dimer (Wardleworth et al, 2002), an ECT2 fragment spanning the YTH domain plus the SLiM-containing proximal part of the IDR, and an m⁶A-containing 10-nt RNA could be modeled. Interestingly, AlphaFold3 generated a model of high confidence overall (Figs. 2B–D and EV1A,B). The model features several interactions between the N8-SLiM and the YTH domain, and situates the SLiM centrally between the YTH domain, the ALBA domains, and the m⁶A-containing RNA (Figs. 2B and EV1A). Because these properties offer straightforward explanations for the reduced ALBA- and RNA-association of ECT2$^{\Delta N8}$ in vivo, we devoted further efforts to the study of the SLiM and refer to it as the YTHDF-ALBA Interaction Motif (YAIM) in the remainder of this report.

## The YAIM is required for ECT2-ALBA interaction and ECT2 function

We next generated a YAIM mutant of ECT2 containing several alanine substitutions (Fig. 2E). The ECT2$^{YAIM}$-mCherry mutant

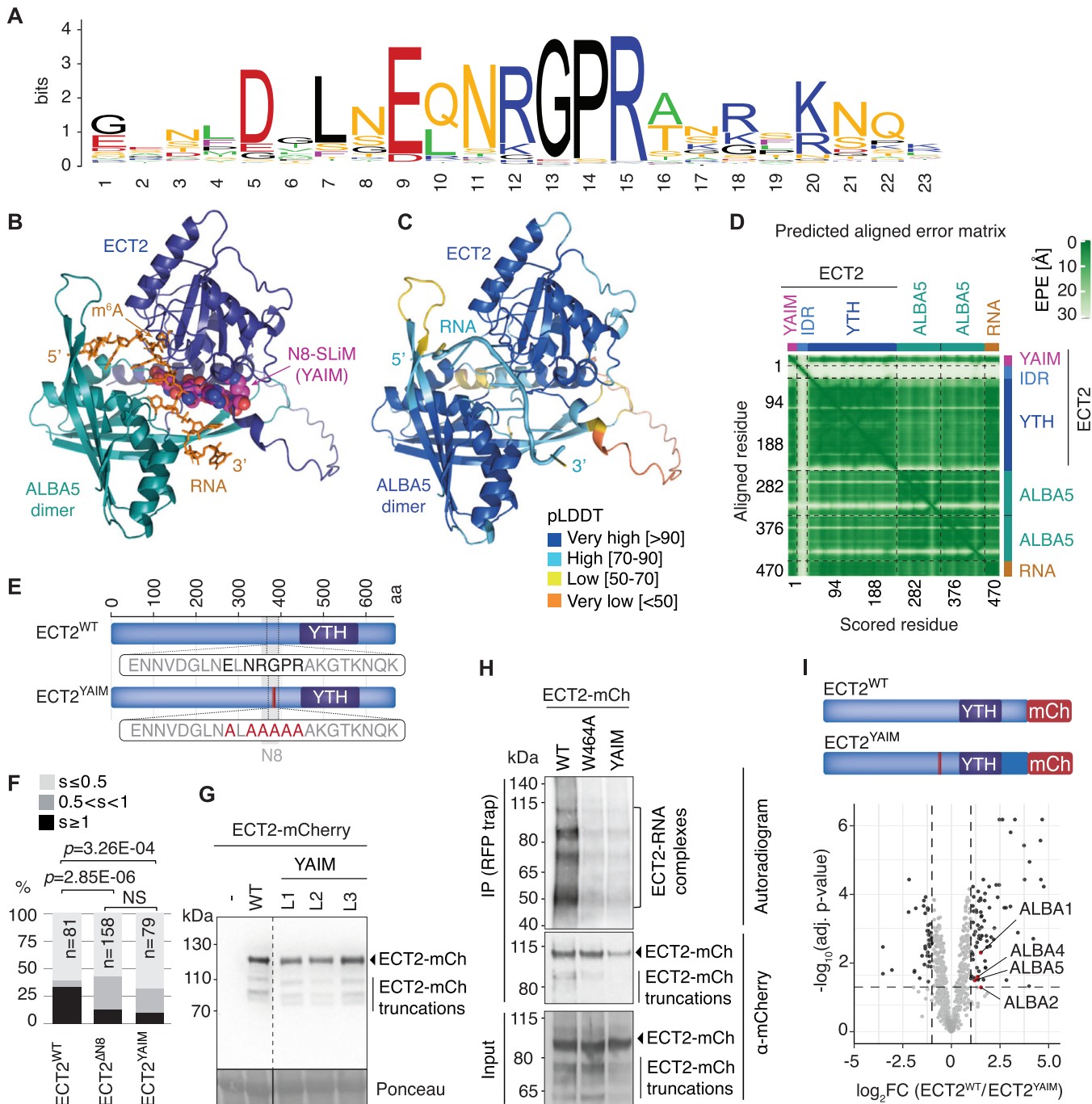

exhibited a reduced *te234* complementation frequency similar to ECT2^ΔN8-mCherry (Fig. 2F), despite the fact that protein levels in several independent transgenic lines were similar to those obtained with an *ECT2^WT-mCherry* transgene (Fig. 2G). These observations demonstrate the in vivo importance of the YAIM for ECT2 function. At the molecular level, the ECT2^YAIM-mCherry mutant also exhibited defects closely resembling those of ECT2^ΔN8-mCherry: less RNA could be crosslinked and immunoprecipitated with ECT2^YAIM-mCherry than with ECT2^WT-mCherry (Fig. 2H), and ALBA1/2/4/5 were depleted in ECT2^YAIM-mCherry immuno-purifications relative to ECT2^WT-mCherry (Figs. 2I and EV1C;

Dataset EV1). We also used the ALBA1 antibody to verify reduced association with ECT2^YAIM-mCherry compared to ECT2^WT-mCherry (Fig. EV1D). Taken together, we conclude that the YAIM is required for ALBA association and for full target RNA-binding of ECT2 in vivo, as predicted by the AlphaFold3 model of the (ALBA5)_2-ECT2-RNA complex. We note, however, that ALBA1/2/4/5 were not the only proteins depleted from ECT2-mCherry purifications upon mutation of the YAIM, perhaps suggesting that the primary function of the YAIM is to mediate ALBA- and RNA-interaction, and that the ECT2-ALBA-RNA complex generates a platform required for interaction with multiple other proteins.

**Figure 2.  The ECT2-ALBA interaction is mediated by a conserved short linear motif in the N8 element of the ECT2 IDR.**

(A) Logo representation of sequence conservation in the N8 region of the IDR of YTHDF proteins in the DF-A, -B, -C, -D, and -E clades of 34 land plant species distributed along all main phylogenetic clades, from liverworts and mosses to flowering plants (Dataset EV7). Fern DF-D proteins and the entire gymnosperm- and fern-specific DF-F clade were excluded from the analysis because they do not have an apparent YAIM motif (see Appendix Fig. S3). The logo (Schneider and Stephens, 1990) was generated using the Weblogo tool (Crooks et al, 2004), and sequences were aligned with MUSCLE (Madeira et al, 2024). (B) AlphaFold3 model of the complex between ECT2 (YTH domain plus a YAIM-containing fragment of the N-terminal IDR), two ALBA5 subunits (ALBA domains only), and a 10-nt RNA [5'-AAA(m$^6$A)CUUCUG-3']. The YAIM is accentuated in space-fill mode (magenta, C; blue, N; red, O), all other protein elements in cartoon mode, and the RNA in stick mode. (C) Same view of the model as in panel (B) but colored according to the predicted local distance difference test (pLDDT) score calculated by Alphafold3 to indicate model confidence on a local per-residue basis (Abramson et al, 2024). (D) 2D plot generated by AlphaFold3 showing the Predicted Aligned Error (PAE) indicating the Expected Position Error (EPE) in Ångströms (white-green scale) in the relative positions of each pair of residues in the complex (Abramson et al, 2024). The location of subunits and structural elements along the axes is indicated. An additional view of the complex is provided in Fig EV1. (E) Schematic representation of the ECT2$^{YAIM}$ mutant with alanine substitutions in the YTH-ALBA Interaction Motif (YAIM) highlighted in red. (F) Categorized leaf size (s) distribution of 9-day-old primary transformants of *te234* mutants expressing wild type or mutant versions of ECT2-mCherry as indicated. Statistical differences between the indicated genotypes were calculated using pairwise Fisher exact tests with Holm-adjusted *p* values. NS not significant. (G) Anti-mCherry immunoblot from total lysates of 9-day-old seedlings of transgenic lines expressing either a fully complementing ECT2$^{WT}$-mCherry transgene (Arribas-Hernández et al, 2018) or the ECT2$^{YAIM}$-mCherry construct (L1-L3, three independent lines), or without any ECT2 transgene (–), all in the *te234* mutant background. Dashed lines indicate that lanes have been removed for presentation purposes. Ponceau staining is used as a loading control. (H) Results of an in vivo UV crosslinking-ECT2-mCherry-immunoprecipitation experiment, followed by PNK-labeling of precipitated RNA with γ-$^{32}$P-ATP. Top panel, autoradiogram of $^{32}$P-radiolabelled RNA-protein complexes purified from plants expressing ECT2$^{WT}$-mCherry, the aromatic cage mutant ECT2$^{W464A}$-mCherry, or ECT2$^{YAIM}$-mCherry. Molecular weight marker positions and the location of the verified ECT2-mCherry-RNA complexes (Arribas-Hernández et al, 2021a) are indicated. The presence of several bands of unequal intensity is due to partial proteolysis of the ECT2 IDR during immunoprecipitation and differential labeling efficiency of the different RNPs (Arribas-Hernández et al, 2021a). Middle and bottom panels, immunoblots against mCherry showing the ECT2-mCherry proteins in the IP (middle) and total lysates (input, bottom). Samples were pools of three independent lines for each genotype. (I) Volcano plot showing a differential abundance of proteins detected by mass spectrometry in mCherry immunoprecipitates from *te234* seedlings expressing either ECT2$^{YAIM}$-mCherry or ECT2$^{WT}$-mCherry. Statistical significance was determined using empirical Bayes statistics with Benjamini–Hochberg adjusted *p* values. The experiment was done in biological triplicates. Source data are available online for this figure.

## A model for concerted m⁶A-ECT-ALBA function in vivo

The results presented so far suggest that ECTs and ALBAs act in concert to bind to m⁶A-sites in mRNA targets. A basic prediction of this hypothesis is that ECT2 and ALBAs are expressed in the same cells. Examination of expression patterns using fluorescent protein fusions expressed under the control of endogenous promoters showed that ALBA1, ALBA2, and ALBA4 are indeed expressed in mitotically active cells of root and leaf primordia, as is ECT2 (Fig. 3A,B). The tight co-expression of ECT2 and ALBA proteins was also evident from analysis of published root single-cell mRNA-seq data (He et al, 2023; Shahan et al, 2022) (Appendix Fig. S4). Further assessment of the subcellular localization by confocal microscopy indicated that ALBA1, ALBA2, ALBA4, and ALBA5 localize to the cytoplasm (Fig. 3C), as do ECT2, ECT3 and ECT4 (Arribas-Hernández et al, 2018; Arribas-Hernández et al, 2021b; Arribas-Hernández et al, 2020).

The model further predicts that ALBAs and ECTs share a significant overlap in mRNA target sets, that they have juxtaposed binding sites around m⁶A sites in those target mRNAs, and that at least some direct mRNA targets associate less with ECTs in vivo in the absence of ALBA proteins. We previously demonstrated the feasibility of using TRIBE (Target Identification of RNA-binding Proteins by Editing) (McMahon et al, 2016) and iCLIP (Individual Nucleotide-Resolution Crosslinking and Immunoprecipitation) (König et al, 2010) to address such predictions using transcriptome-wide analyses in vivo (Arribas-Hernández et al, 2021a; Arribas-Hernández et al, 2021b; Meyer et al, 2017). In TRIBE, the catalytic domain (cd) of the A-I RNA-editing enzyme ADAR is fused to the RNA-binding protein of interest, and targets are identified by mRNA-seq as mRNAs containing sites significantly more edited in cells expressing the ADAR$_{cd}$ fusion compared to a free ADAR$_{cd}$ control (Arribas-Hernández et al, 2021a; McMahon et al, 2016). TRIBE can also be used to estimate differential protein-mRNA association between two conditions based on quantitative changes in editing proportions in target mRNAs. For example, many shared ECT2/3 target mRNAs are more highly edited by ECT3-ADAR$_{cd}$ in the absence of ECT2, indicating that the two proteins compete for the same binding sites in vivo (Arribas-Hernández et al, 2021b). In iCLIP, target mRNAs are identified by co-purification with the protein of interest after covalent crosslinking in vivo, and binding sites are deduced from the position of frequent reverse transcription termination events at crosslink sites (König et al, 2010). We, therefore, set out to test predictions on shared and interdependent ECT-ALBA target binding in vivo using combined iCLIP and TRIBE analyses focused on ALBA4 (long form), ALBA2 (short form), and ECT2.

## Identification of mRNA targets of ALBA4 using iCLIP

We first aimed to identify direct mRNA targets and binding sites of ALBA4 via iCLIP. To this end, we used transgenic lines expressing *ALBA4-GFP* under the control of the endogenous *ALBA4* promoter in the *alba4-1 alba5-1 alba6-1* (henceforth, *alba456*) mutant background (Appendix Fig. S5A,B), verified to carry T-DNA-induced knockout mutations in all three *ALBA* genes by RT-qPCR (Appendix Fig. S5C) and western blot (Appendix Fig. S5D) analyses. Initial immunoprecipitation tests with or without prior UV crosslinking and followed by polynucleotide kinase (PNK) labeling established that RNA-protein complexes were specifically purified with ALBA4-GFP after UV crosslinking (Fig. 4A). We therefore prepared and sequenced libraries from RNA immuno-purified with ALBA4-GFP or GFP alone after crosslinking in vivo (Appendix Fig. S6A–D), using the recently developed iCLIP2 protocol (Buchbender et al, 2020; Lewinski et al, 2024). This effort identified 379,670 high-confidence replicated sites for ALBA4-GFP, corresponding to 7744 genes (henceforth referred to as ALBA4 iCLIP2 targets). We further defined a "strong" set by filtering low scores, resulting in 63,695 sites mapping to 7509 genes. In the GFP-only samples, only 81 sites in 13 genes were detected (Fig. 4B; Dataset EV2). Thus, nearly all ALBA4 iCLIP2 targets are strong candidates for bona fide ALBA4 target mRNAs.

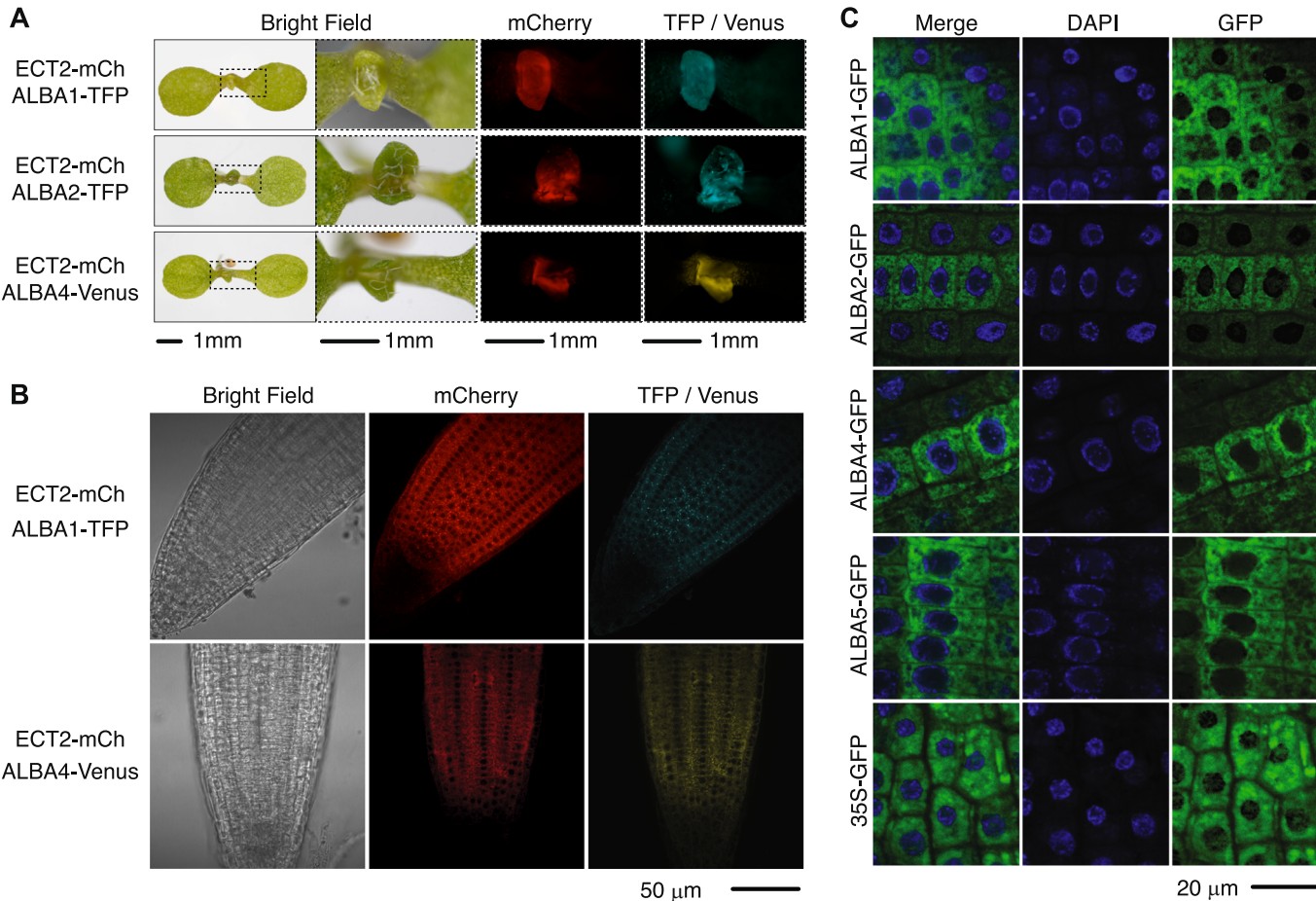

**Figure 3.   The expression patterns and subcellular localizations of ECTs and ALBAs overlap.**

(**A**) Fluorescence microscopy of 5-day-old seedlings co-expressing ECT2-mCherry and ALBA1-TFP (top panel), ECT2-mCherry and ALBA2-TFP (middle panel), or ECT2-mCherry and ALBA4-Venus (bottom panel). (**B**) Confocal microscopy images of mCherry, TFP and Venus fluorescence in root tips of plants co-expressing ECT2-mCherry and ALBA1-TFP (top) or ECT2-mCherry and ALBA4-Venus (bottom). (**C**) Confocal images of GFP fluorescence and DAPI staining in root tips of plants expressing ALBA1-GFP, ALBA2-GFP, ALBA4-GFP, ALBA5-GFP, and 35S-GFP. Source data are available online for this figure.

## mRNA target sets of ALBA proteins overlap significantly with those of ECT2/ECT3

We first noticed that ALBA4 iCLIP2 sites occurred in coding regions and, even more predominantly, in 3′-UTRs, with the 3′-UTR enrichment particularly apparent in the strong set (Fig. 4C). Importantly, more than 90% of ECT2 iCLIP targets are also ALBA4 iCLIP2 targets (Fig. 4D). Hence, ECT2 mainly binds to mRNAs that are also targeted by ALBA4. To corroborate this essential conclusion, we employed TRIBE to identify targets of both a long (ALBA4) and a short (ALBA2) ALBA protein family member by independent means. We used the improved variant HyperTRIBE relying on a hyperactive mutant (E488Q) of the $ADAR_{cd}$ (Xu et al, 2018) for ALBA2, but had to proceed with TRIBE for ALBA4, because expression of the hyperactive $ADAR_{cd}$ fused to ALBA4 was lethal (see Methods). In both cases, lines expressing comparable levels of free and ALBA-fused $ADAR_{cd}$ were selected for mRNA-seq analysis (Appendix Fig. S7). Significantly differentially edited sites between fused and free $ADAR_{cd}$ exhibited higher editing proportions in the ALBA2/4-FLAG-$ADAR_{cd}$ fusions, as expected (Fig. 4E,F). These differentially edited sites defined 5272 target

mRNAs for ALBA2 and 5995 for ALBA4 (Fig. 4E,F; Dataset EV3). Using these target sets, ALBA4 iCLIP2 targets and the previously defined ECT2/3 targets (Arribas-Hernández et al, 2021a; Arribas-Hernández et al, 2021b) for comparative analyses, we revealed the following three properties of ALBA2/4 and ECT2/3 target mRNAs and the relations between them. (1) The ALBA4 iCLIP2 target set is robust, because the overlap with ALBA4-TRIBE is significant (Fig. 4G). In particular, TRIBE support of ALBA4 iCLIP2 targets is prominent for those target mRNAs with multiple called iCLIP peaks (Fig. EV2A,B). (2) ALBA4 and ALBA2 target a common set of mRNAs (Fig. 4H) and differences between the two target sets can largely be explained by the tissue source used for the analysis (aerial tissues for ALBA2, roots for ALBA4) (Fig. EV2C). (3) The overlaps between the ECT2/3 target set and both the high-confidence set of ALBA4 targets supported by iCLIP2 and TRIBE and the set of ALBA2 HyperTRIBE targets are highly significant, as demonstrated by comparison to corresponding random target sets (Figs. 4I,J and EV2D–F; Appendix Fig. S8; Dataset EV4). We conclude that ECT2/3 and ALBA2/4 mRNA target sets significantly overlap, thus fulfilling a second key requirement of the model of concerted mRNA binding by ECT-ALBA modules.

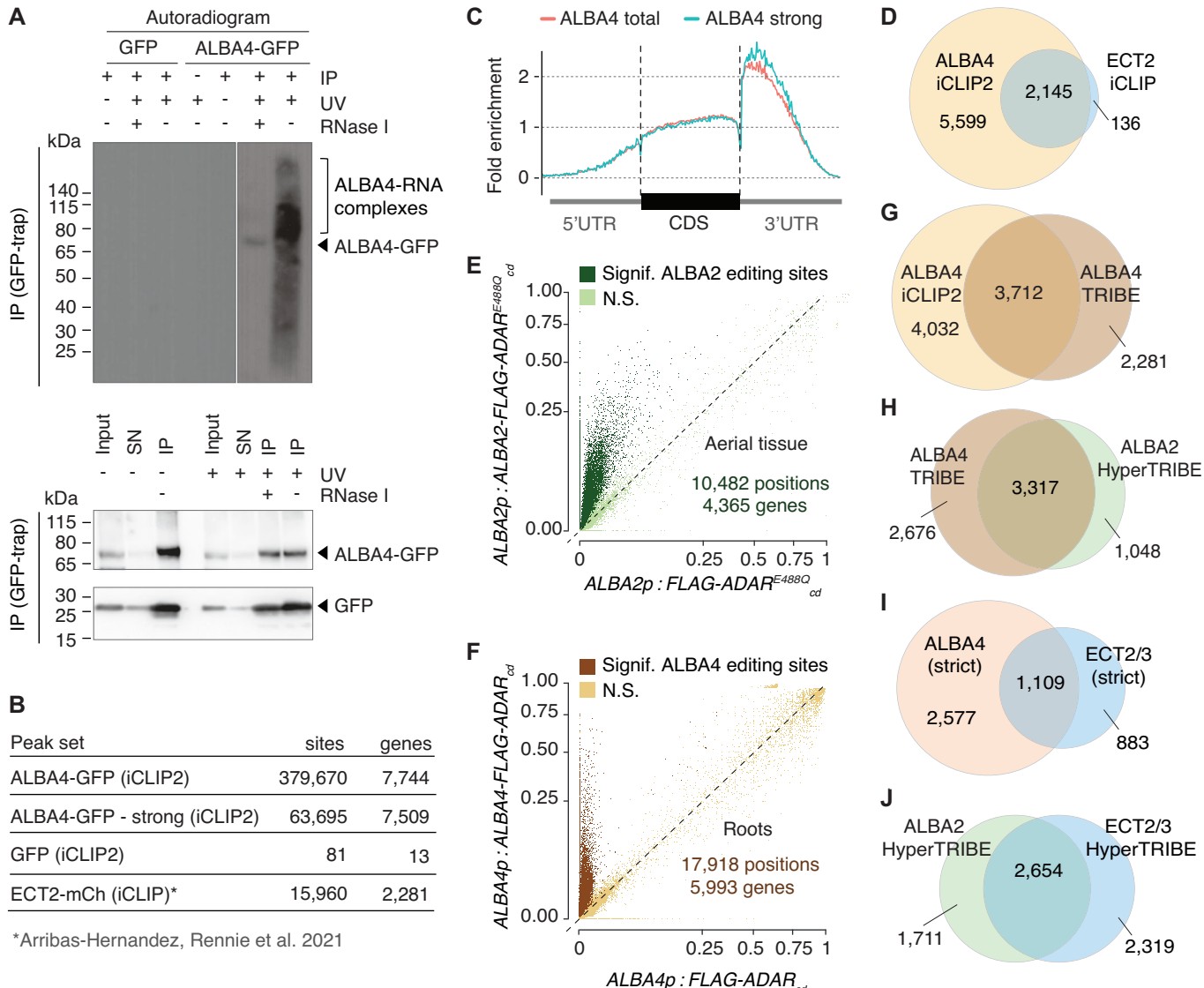

**Figure 4. The mRNA targets bound by ECT2/3 and ALBA2/4 overlap substantially.**

(A) Top, autoradiogram of $^{32}$P-labeled RNA-protein complexes obtained by PNK/γ-$^{32}$P-ATP labeling of immunopurified material from ALBA4-GFP- or GFP-expressing plants. Immunoprecipitations were carried out with or without UV crosslinking and after precipitation with GFP-Trap beads (IP+). (IP−) indicates mock immunoprecipitation with RFP-Trap beads. Treatment of the precipitate with RNase I (+RNase) indicates the size of the precipitated protein. Marker positions and the location of the ALBA4-GFP RNA adducts are indicated. Bottom, immunoblots of input, supernatant (SN) after IP, and immunoprecipitated (IP) fractions, probed with GFP antibodies. Samples are pools of 3 independent lines for each genotype. (B) Number of called iCLIP peaks and associated genes for ALBA4-GFP, GFP alone, and ECT2-mCherry (Arribas-Hernández et al, 2021a). Strong ALBA4-GFP peaks are defined as those with a score higher than the median, per gene. (C) Scaled metagene profiles showing the enrichment along the gene body (5'UTR, CDS or 3'UTR) of ALBA4-GFP iCLIP2 peaks. (D) Overlap of ECT2 and ALBA4 iCLIP mRNAs. The overlap is highly significant ($p = 0$ according to permutation test based on random sampling of genes from transcriptome with matched expression patterns, see Methods). (E) Scatter plot of the editing proportions (E.P. = G/(A + G)) of potential and significant editing sites (E.S.) determined by comparing mRNA-seq data obtained from transgenic lines expressing ALBA2-FLAG-ADAR$_{cd}$$^{E488Q}$ or FLAG-ADAR$_{cd}$$^{E488Q}$ in the Col-0 background, both under the control of the ALBA2 promoter (seedlings, shoot tissue). Significance was determined using the hyperTRIBER pipeline (Rennie et al, 2021), specifying an adjusted $p$ value <0.01 and log$_2$ fold-change >1. (F) The same analysis as in (E), but carried out with roots of lines expressing ALBA4-FLAG-ADAR$_{cd}$ or FLAG-ADAR$_{cd}$ under the control of the ALBA4 promoter in the Col-0 background. (G) Overlap of ALBA4 targets identified using iCLIP2 and TRIBE analysis. The overlap is highly significant ($p = 0$, permutation test, as in D). (H) Overlap between ALBA4-TRIBE targets (roots) and ALBA2 HyperTRIBE targets (shoots). The overlap is highly significant ($p = 0$, permutation test, as in D). Most non-overlapping targets are expressed specifically in shoots or roots (Fig. EV2C). (I) Overlap between high-confidence ALBA4 targets, supported by iCLIP and TRIBE, and ECT2/3 targets, supported by ECT2/3 HyperTRIBE and ECT2 iCLIP. The overlap is highly significant ($p = 0$, permutation test, as in D). (J) Overlap between ALBA2 HyperTRIBE targets and ECT2/3 HyperTRIBE targets. The overlap is highly significant ($p = 0$, as in D). Source data are available online for this figure.

## ALBA proteins bind to pyrimidine-rich elements in the vicinity of m⁶A

We next analyzed the positions of ALBA4 binding sites in their targets using the iCLIP2 data. Metagene analysis normalizing for region length showed a peak in the density of ALBA4 binding sites in 3′-UTRs, if less pronounced than ECT2 binding sites and m⁶A-sites because ALBA4 binding sites also occur in coding regions as noted above (Fig. 5A). The *RPS7A* and *TUBULIN ALPHA-5* genes provide illustrative examples of this close alignment of m⁶A, ECT2 and ALBA4 sites (Fig. 5B). Both ALBA4 iCLIP2 and ECT2 iCLIP peaks (Arribas-Hernández et al, 2021a) are enriched upstream of m⁶A sites determined by Nanopore direct RNA sequencing (Parker et al, 2020) (Fig. 5C,D), with ALBA peaks situated either at or slightly upstream of ECT2 peaks (Fig. 5E). Strikingly, the enrichment of ALBA4 peaks at m⁶A-sites was much more pronounced when considering peaks in ECT2 targets compared to non-targets. Indeed, the ALBA4 peak enrichment around m⁶A-sites in ECT2 non-targets showed a distribution similar to the location-matched background (Fig. 5F). These key observations demonstrate that the important prediction of a juxtaposition of ECT2 and ALBA4 binding sites on target mRNAs is fulfilled, and strongly suggest mutual dependence on target mRNA binding.

Because we previously showed that several sequence motifs are enriched around ECT2 binding sites (Arribas-Hernández et al, 2021a), we went on to study whether any of these motifs were enriched at ALBA4 binding sites. We included 6 motifs identified as enriched around ECT2 iCLIP sites in our previous study (Arribas-Hernández et al, 2021a). This analysis revealed that uridine- or pyrimidine-rich motifs in the immediate vicinity of m⁶A/ECT2 binding sites are strongly enriched precisely at ALBA4 crosslink sites (Fig. 5G), suggesting that these sequences may be ALBA4 binding sites in vivo.

## Deep learning supports pyrimidine-rich elements in the vicinity of m⁶A as determinants of ALBA4-ECT2 binding

One potential pitfall of this conclusion is that the photochemical properties of nucleobases result in a bias of UV-induced RNA-protein crosslinks to occur at uridines (Angelov et al, 2023; Hafner et al, 2021) such that iCLIP sites can be located at nearby uridines if the actual binding site lacks this nucleotide. For example, many miCLIP sites obtained by UV crosslinking of an m⁶A-specific antibody to RNA in vitro map to uridines surrounding the uridine-depleted major m⁶A consensus site (DRACH) (Arribas-Hernández et al, 2021a). Therefore, we employed neural networks to identify sequence elements that distinguish m⁶A sites bound by ECT2/ALBA4 from m⁶A sites not bound by these proteins. We first collected Arabidopsis m⁶A sites from multiple published sources and curated a compendium of 41,883 non-overlapping m⁶A sites which have properties highly consistent with the smaller set of sites identified by Nanopore direct RNA sequencing (Parker et al, 2020) (Fig. EV3A–F; Dataset EV5, see Methods). The high quality of these sites is supported by their strong enrichment in single-nucleotide resolution m⁶A sites recently obtained with the m⁶A-specific allyl chemical labeling and sequencing method (m⁶A-SAC-seq (Wang et al, 2024)) (Fig. EV3G). Of these, 16,406 sites were annotated as ECT2-positive and 22,866 were ALBA4-positive (Fig. 6A). Although there was a large overlap between the two proteins, there was a sizable set of bound sites unique to each protein (Fig. EV3H),

allowing analysis of sequence features of sites bound by both proteins and uniquely bound sites. We then used sequences surrounding all sites for input into a neural network trained simultaneously on two binary outputs: whether ECT2 was bound or unbound, and whether ALBA4 was bound or unbound (Fig. 6A). This model performed well when predicting the presence of ALBA4 or ECT2 at m⁶A sites on gene sets excluded during model training ("held-out set"; average AUC = 0.74 (ECT2) and 0.76 (ALBA4), based on fivefold cross validation), with predicted binding probabilities clearly distinguishing between bound and unbound sites (Fig. 6B). As expected, predicted binding probabilities for the two proteins correlated (PCC = 0.71, Fig. 6C). Importantly, some differences between the two suggested that the model had learned specific sequence patterns relevant to each protein. To investigate this, we leveraged the filters learned in the first convolutional layer, since these represent motifs identified de novo by the model. We converted the sequences of the highest-scoring instances into position weight matrices and fit a generalized linear model predicting motif presence additively from the network-predicted ECT2 and ALBA4 binding probabilities (see Methods). From this model, the coefficient for each protein (motif score) can be interpreted as the effect of that protein controlling for the other (Fig. 6D). This analysis identified the uridine-/pyrimidine-rich motifs UAUUUU and UUUACUUU as determinants of both ECT2-bound and ALBA4-bound m⁶A sites (Fig. 6D). Indeed, the UAUUUU and UUUACUUU motifs were highly enriched at ALBA4 iCLIP sites and located just upstream of ECT2 iCLIP sites (Fig. 6E), thus providing independent experimental evidence that these motifs act as ALBA4 binding sites. This conclusion is particularly important because it provides a simple molecular explanation for our previous machine learning-based finding that uridine- or pyrimidine-rich motifs are important for the distinction between m⁶A sites bound or not by ECT2 (Arribas-Hernández et al, 2021a): juxtaposed m⁶A sites and uridine-/pyrimidine-rich elements provide the context required for binding of the ECT-ALBA module.

## Binding to target mRNA in vivo involves mutual ALBA-ECT dependence

We next assessed whether ALBA proteins are necessary for mRNA target association of the wild type ECT2 protein. Initially, we used the CLIP-PNK assay with ECT2-mCherry expressed in wild type, or the *alba1-2 alba2-2 alba4-1 alba5-1* (henceforth *alba1245*) or *alba456* mutant backgrounds, carrying T-DNA insertion alleles in the corresponding *ALBA* genes (Appendix Figs. S2B and S5, see Methods). These experiments showed that ECT2-mCherry was associated with less RNA in the *alba* mutants compared to wild type, with the clearest effects (~2.5-fold reduction) observed in *alba456* mutants (Figs. 7A and EV4A). We next used ECT2 HyperTRIBE to estimate the relative target mRNA binding in wild type and in *alba1245* mutants by differential editing. We chose this method both to gain sensitivity and to assess directly whether mRNAs that associate less with ECT2 in vivo in *alba1245* mutants are, in fact, dual ECT2/ALBA targets. We selected five independent lines expressing ECT2-ADAR in both wild type and *alba1245*, and performed mRNA-seq of root tissues to provide the raw data for analysis of differential editing. Positions exhibiting significant differential editing according to the hyperTRIBER package (Rennie et al, 2021) were strongly biased in the direction of lower editing in *alba1245*, although these results were potentially biased by the

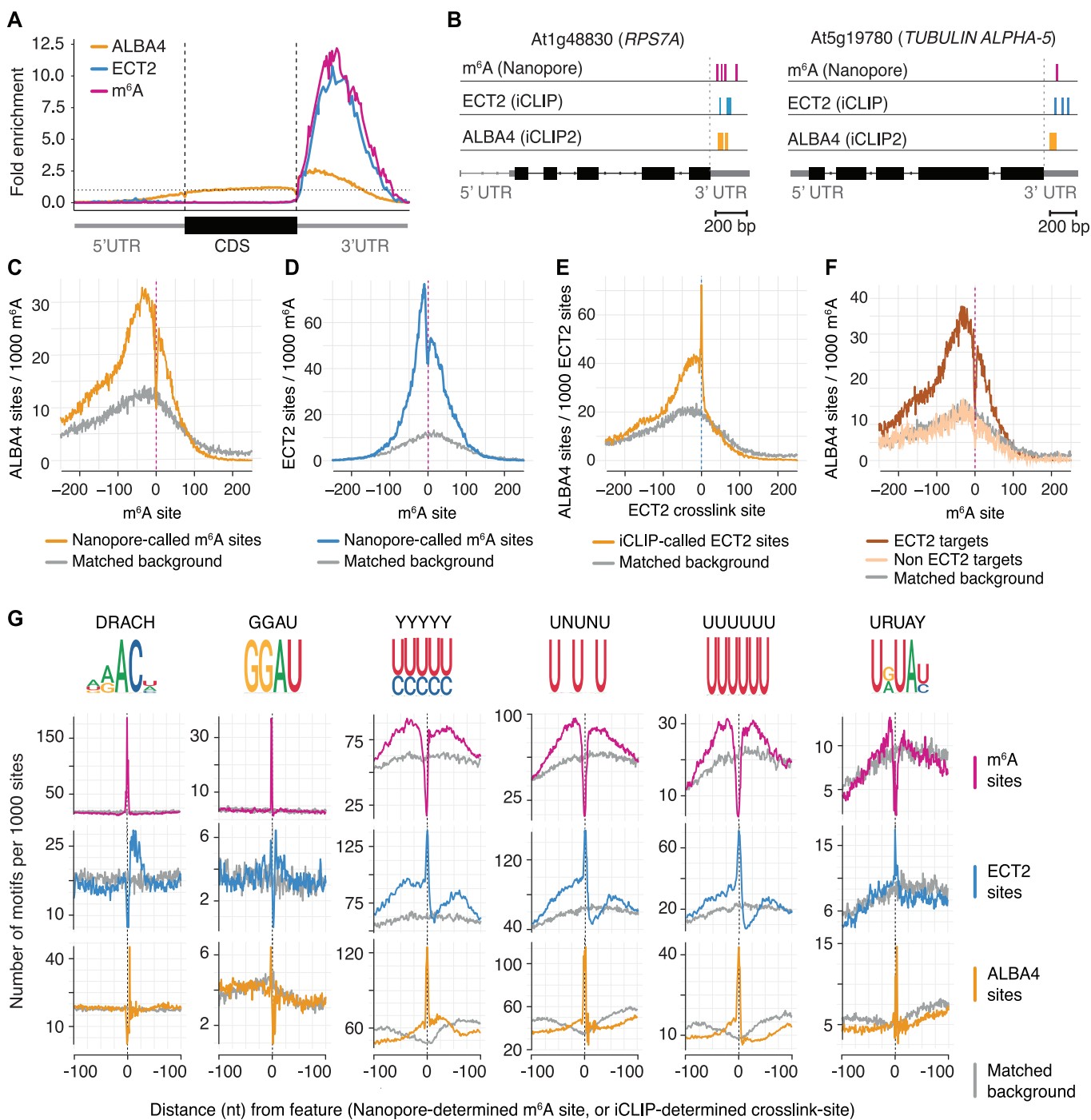

Figure 5. ALBA4 binds to pyrimidine-rich elements juxtaposed to m6A.

(A) Scaled metagene profiles showing the enrichment along the gene body (5'UTR, CDS, or 3'UTR) of the called ALBA4 iCLIP2 peaks. ECT2 iCLIP peaks (Arribas-Hernández et al, 2021a) and Nanopore-determined m6A density (Parker et al, 2020) are shown for reference. (B) Representative examples of ECT2 and ALBA4 common targets showing the location of ALBA4 iCLIP2 and ECT2 iCLIP crosslink sites (Arribas-Hernández et al, 2021a), and m6A sites (Parker et al, 2020). (C) Number of ALBA4 iCLIP2 crosslink sites per 1000 Nanopore-derived m6A sites, as a function of distance from the m6A sites. (D) Number of ECT2 iCLIP crosslink sites per 1000 Nanopore-derived m6A sites, as a function of distance from the m6A sites. (E) Number of ALBA4 iCLIP2 crosslink sites per 1000 ECT2 crosslink sites, as a function of distance from the ECT2 crosslink sites. (F) Number of ALBA4 iCLIP2 crosslink sites per 1000 nanopore-derived m6A sites, as a function of distance from the m6A site and according to whether containing genes are also targets of ECT2 or non-ECT2 targets. For each set, a matched background set was defined as positions on similarly expressed genes with a similar metagene distribution to the true set. (G) Number of the indicated motifs (selected from (Arribas-Hernández et al, 2021a)) per 1000 nanopore-determined m6A sites (top), ECT2 iCLIP crosslink sites (middle) or ALBA4 iCLIP2 crosslink sites (bottom). For each set, a matched background set was defined as positions on similarly expressed genes with a similar metagene distribution to the true set.

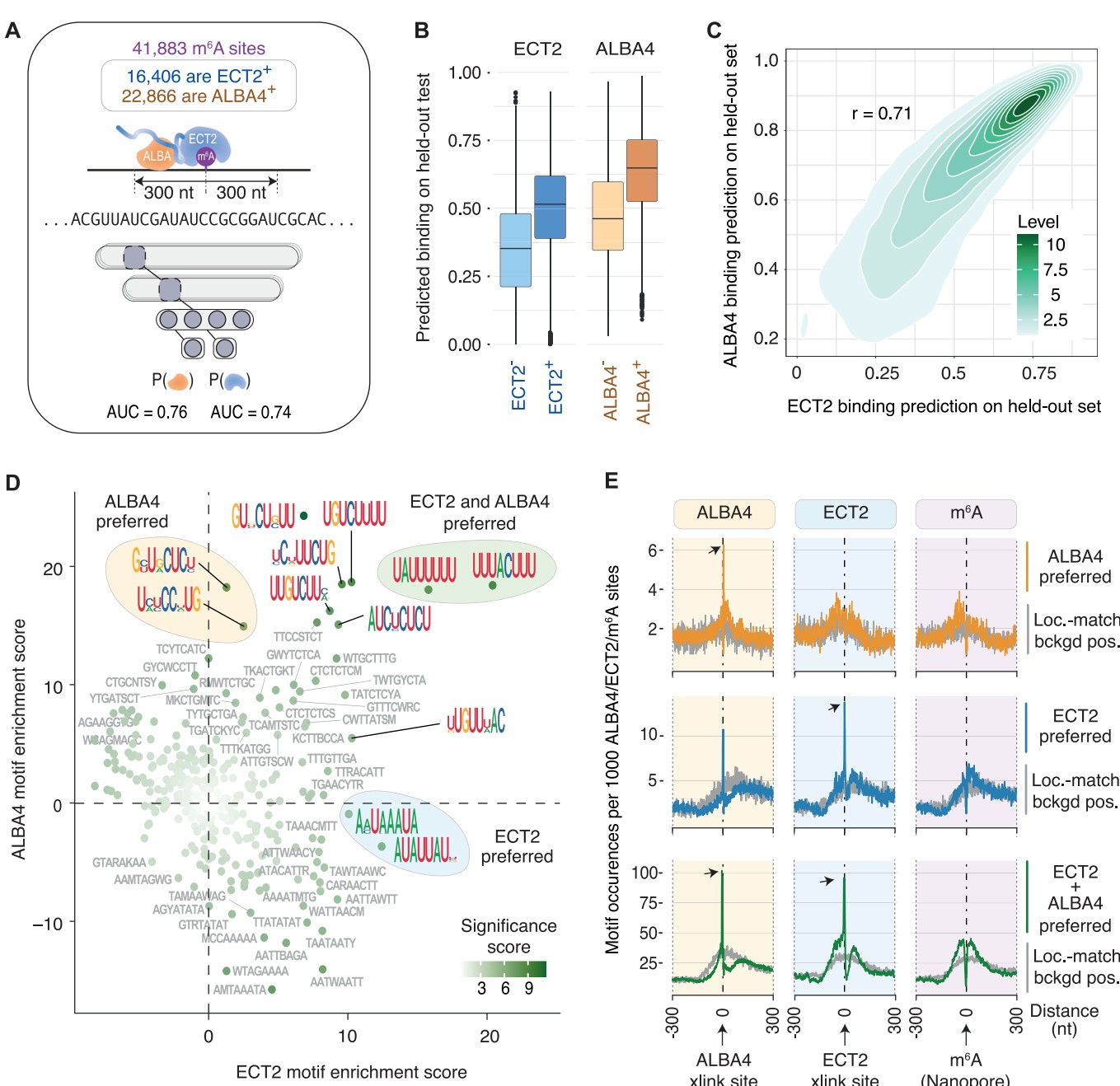

**Figure 6. Neural network analysis identifies U-rich motifs in the vicinity of m⁶A as determinants of ALBA4-ECT2 binding.**

(A) Strategy for deep learning. m⁶A sites were annotated according to the presence or absence of either ECT2 or ALBA4, and a convolutional neural network was trained, which takes sequences surrounding m⁶A as input and predicts the probability of ECT2 and ALBA4 binding. (B) Boxplots showing predicted binding probabilities from the network, split according to protein and binding status. From left to right: n = 25,477, 16,406, 19,017, and 22,866, and boxplots show {minima, 25th percentile, median, 75th percentile, maxima}. (C) Scatter plot of the predicted ALBA4 binding probabilities against the ECT2 binding probabilities from the network. Counts depict the density of sites. (D) Output-specific enrichment scores for de novo motifs learned by the convolutional neural network, calculated using a generalized linear model for predicting motif presence from the predicted presence of ECT2 and ALBA4 at m⁶A-centered sequences using model. Colored circles indicate interesting motifs determined as specific to ALBA4 (yellow), ECT2 (blue), or both (green). (E) Enrichment of motif sets indicated in D around ALBA4 iCLIP2, ECT2 iCLIP, and nanopore-derived m⁶A sites (Parker et al, 2020). Gray shows location-matched background positions.

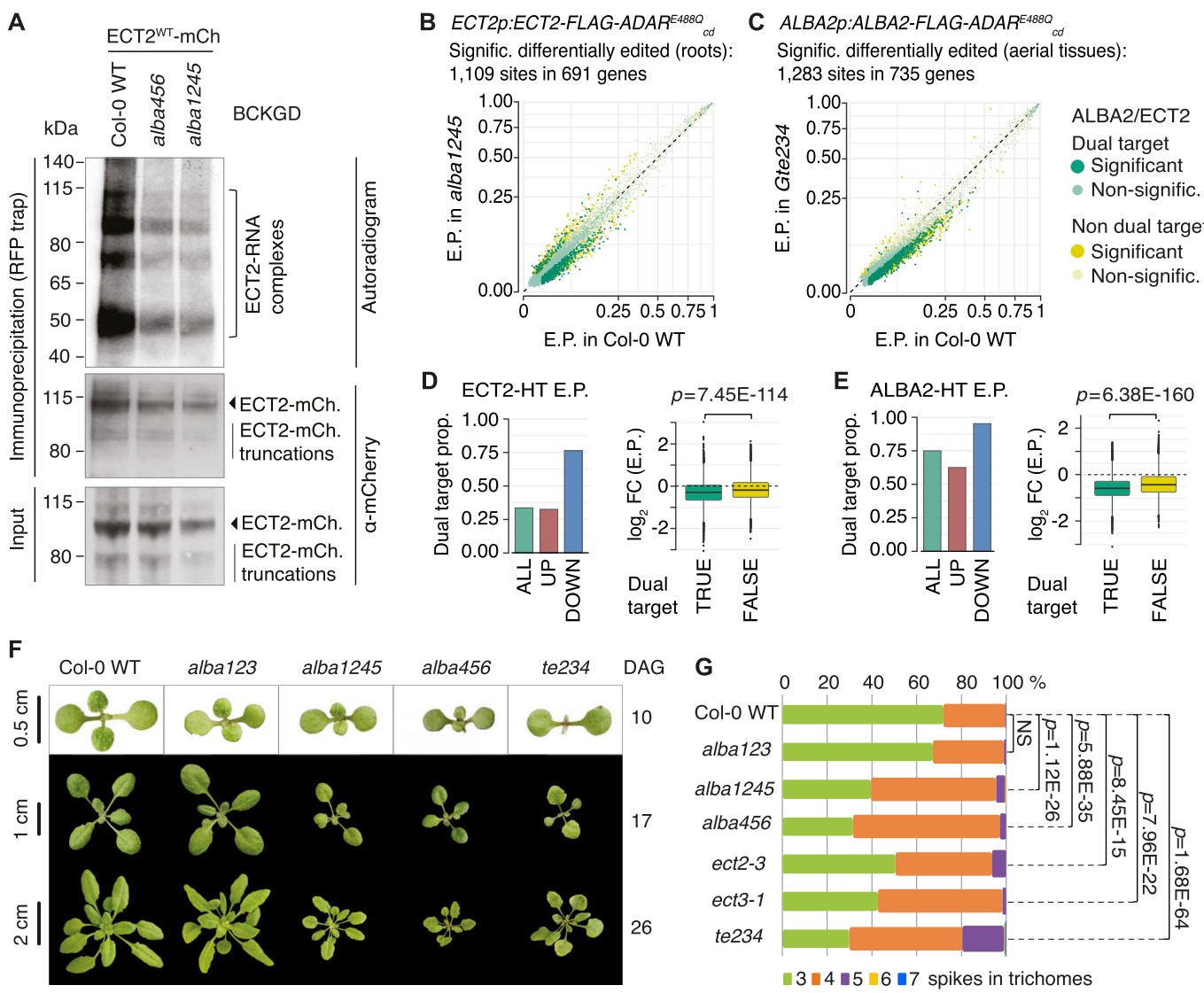

**Figure 7. ALBA proteins are required for ECT2 target mRNA binding and biological function.**

(A) Results of an in vivo UV crosslinking-ECT2-mCherry immunoprecipitation experiment, followed by PNK-labeling of precipitated RNA with γ-$^{32}$P-ATP. Top panel, autoradiogram of $^{32}$P-radiolabelled RNA-protein complexes purified from plants expressing ECT2$^{WT}$-mCherry in the indicated genetic backgrounds. Molecular weight marker positions and the location of the verified ECT2-mCherry-RNA complexes (Arribas-Hernández et al, 2021a) are indicated. The presence of several bands of unequal intensity is due to partial proteolysis of the ECT2 IDR during immunoprecipitation, and differential labeling efficiency of the different RNPs (Arribas-Hernández et al, 2021a). Middle and bottom panels, mCherry immunoblots of the immunoprecipitated (middle) and total fractions (input, bottom). Samples were pools of three independent lines for each genotype. (B) Scatter plot showing the editing proportions (E.P. = G/(A+G)) of ECT2-FLAG-ADAR$_{cd}^{E488Q}$-catalyzed editing sites between Col-0 WT and *alba1245*. Green, sites whose change in editing proportions is statistically significant and that are located in dual-bound mRNAs. Yellow, sites whose change in editing proportions is statistically significant but that are located in mRNAs not targeted by both ECT2 and ALBA4 (non-dual bound). Light green/light yellow, candidate sites whose change in editing proportions is not statistically significant. (C) Scatter plot showing the editing proportions of ALBA2-FLAG-ADAR$_{cd}^{E488Q}$-catalyzed editing sites between Col-0 WT and *ect2-3 ect3-2 ect4-2* (*Gte234*). Color scheme as in (B). (D) Quantification of the tendency of sites differentially edited by ECT2-FLAG-ADAR$_{cd}^{E488Q}$ between Col-0 and *alba1245* to be less highly edited in *alba1245*. Left, histogram showing the fraction that sites in dual-bound targets comprise of either less highly edited sites in *alba1245* (down) or more highly edited sites in *alba1245* (up). The histogram also illustrates the fraction of editing sites in dual-bound targets relative to all editing sites for comparison. Right, boxplot showing the median log$_2$ differential editing proportions for editing sites either in dual-bound mRNA targets (TRUE, $n = 10,272$) or in other mRNAs (FALSE, $n = 20,343$), indicated $p$ value is based on two-sample $t$-test and boxplots show {minima, 25th percentile, median, 75th percentile, maxima}. (E) Quantification of the tendency of sites differentially edited by ALBA2-FLAG-ADAR$_{cd}^{E488Q}$ between Col-0 and *Gte234* to be less highly edited in *Gte234*. Analogous to the analyses presented in (D) for ECT2-ADAR in Col-0 vs *alba1245* ($n = 8248$ and $n = 20,024$ for dual-bound and other mRNAs, respectively). (F) Representative photographs of seedlings and rosettes of the indicated genotypes at three different time points were given in days after germination (DAG) in soil. (G) Trichome branching is sorted by the number of spikes in the indicated genotypes. Branches were counted on at least 150 trichomes on each of at least six plants for each genotype ($n = \sim1000$). Data were fitted to a proportional odds model in R for statistical analyses (see Methods) with Bonferroni-corrected $p$ values. NS not significant. Source data are available online for this figure.

expression of the ECT2-ADAR fusion protein not being balanced between the two conditions (Fig. EV4B,C,H). For this reason, we developed a highly robust alternative statistical modeling approach, correcting the editing proportions for mRNA levels of ADAR to obtain a smaller, high-confidence set of significantly differentially edited sites between the two backgrounds (see Methods). As a further control, we also performed differential editing analysis using only those replicates whose ECT2-ADAR expression was nearly perfectly matched as judged by both mRNA-seq read densities and protein blots, resulting in a smaller set of sites which overlapped significantly with the set from the robust modeling approach (Fig. EV4F,G). Overall, these analyses converged on the same conclusion: editing proportions by ECT2-FLAG-ADAR$_{cd}^{E488Q}$ in ECT2/ALBA4 dual mRNA targets tended to be higher in wild type than in *alba1245* mutants, indicating that ALBA proteins facilitate target mRNA binding of ECT2 in vivo (Fig. 7B,D). Because the structural model of the ALBA-ECT2 interaction suggests that RNA association by the ALBA domain may also be enhanced by ECT proteins, we did the reciprocal experiment with the short ALBA2 protein. Thus, we expressed ALBA2-FLAG-ADAR$_{cd}^{E488Q}$ either in wild type or *ect2-3 ect3-2 ect4-2* (*Gte234*) mutant backgrounds and carried out analysis of differential editing proportions as above. We found that editing proportions by ALBA2-FLAG-ADAR$_{cd}^{E488Q}$ of ALBA2-ECT2 dual targets were higher in wild type than in *Gte234* mutants (Figs. 7C,E and EV4D,E,I), indicating that there is mutual ALBA-ECT dependence for mRNA target association in vivo. Taken together, our CLIP-PNK assays and TRIBE-based assessment of target mRNA association in vivo support the conclusion that the ALBA domain acts as a unit with the YTH domain to facilitate m6A-binding.

### Inactivation of ALBA and ECT genes causes similar developmental phenotypes

We finally characterized mutants in *ALBA* genes to assess whether they exhibit phenotypes characteristic of reduced m6A-ECT function. As previously reported, single *alba* mutants (Appendix Fig. S2) did not show obvious developmental phenotypes (Tong et al, 2022). In contrast, *alba123* mutants and, in particular, *alba1245* and *alba456* mutants showed pleiotropic developmental defects, including slower growth, defects in leaf morphology, and delayed flowering (Figs. 7F and EV5A,B). Similar observations on the smaller stature of *alba456* mutants have been reported by others (Tong et al, 2022). Although some of these phenotypes are reminiscent of phenotypes displayed by *ect2 ect3 ect4* mutants, they are not identical. We, therefore, assessed a quantifiable phenotype seen consistently in mutants with defects in m6A-ECT function: increased branching of leaf epidermal hairs (trichomes, (Bodi et al, 2012)) that can be detected even in single *ect2* and *ect3* mutants (Arribas-Hernández et al, 2018; Scutenaire et al, 2018; Wei et al, 2018). We found that *alba1245* and *alba456* mutants showed increased trichome branching (Fig. 7G), with a phenotypic strength intermediate between *ect2* or *ect3* single mutants and the *te234* triple knockout mutant. Importantly, ECT2 protein levels in *alba1245* and *alba456* mutants were only slightly lower than in wild type (Fig. EV5C), excluding the trivial possibility that the phenotypic similarity between composite *alba* and *ect* mutants is due to drastically reduced ECT protein levels in *alba1245* and *alba456*. We conclude that the developmental defects of composite

*alba* mutants are consistent with defective m6A-ECT function, as predicted by the model of m6A-ECT interaction facilitated by ALBA proteins.

## Discussion

Our results on the ALBA-ECT interaction and target binding in vivo provide strong support for the conclusion that the YTH domain of major plant YTHDF proteins is insufficient for full m6A binding in vivo, because it requires facilitation by ALBA proteins. In the following paragraphs, we discuss how this new understanding of the m6A-YTH interaction impacts the thinking of m6A-mediated genetic control in plants and other eukaryotes.

### Functional implications of recognition of m6A by the ALBA-YTHDF module

The discovery that m6A reading in plants involves YTHDF-m6A binding modulated by a third player, the ALBA proteins, introduces increased potential to integrate information into combinatorial control of biological effects of m6A. A key determinant of those effects is the fraction of m6A target mRNAs bound by YTHDF, in turn, determined by the stoichiometry of m6A modification in mRNA, and YTHDF concentration and affinity for m6A-sites. Since we now understand that the affinity is not a constant, but must be tunable via, for instance, ALBA concentration and post-translational modification, we envision that plants have evolved to take advantage of this combinatorial potential to generate a gradient of m6A outputs that matches the cellular environment measured by multiple environmental and developmental sensors.

### Conservation of the ALBA-YTHDF unit and generality of RBP-assisted m6A-YTH interaction

It is an important observation that Arabidopsis YTHDF proteins both with and without the molecular properties required to complement organogenesis defects of *te234* mutants (Flores-Téllez et al, 2023) retain the conserved YAIM and interact with ALBA4. This observation further supports the generality of ALBA-assisted m6A-binding among Arabidopsis YTHDF proteins. Thus, it is a pertinent question how widespread this phenomenon is. The YAIM is deeply conserved in land plant YTHDF proteins, strongly suggesting that the ALBA-YTHDF unit is conserved over the 500 million years of land plant evolution. In fungi and animals, the YAIM is not conserved, and fungal and animal ALBA-family proteins are so divergent in sequence that conservation of the details of their molecular functions cannot be assumed. In addition, *Trypanosoma brucei* where ALBA proteins clearly perform functions in mRNA control (Bevkal et al, 2021; Mani et al, 2011) does not encode YTHDF proteins, providing an example that the two families do not always have linked functions in eukaryotes. These observations raise two immediate questions.

First, given the deep conservation of the YTH domain, it is of interest how the m6A-YTHDF interaction is made efficient in organisms where ALBA proteins are unlikely to assist binding directly as in plants. We see two possible answers. Either other, as yet unidentified classes of RBPs evolved to facilitate m6A reading by YTHDF proteins, or the YTHDF proteins evolved to read m6A

independently of other RBPs. In the latter case, comparative structure-function studies between, for instance, Arabidopsis and human YTHDF-m6A-RNA interactions should reveal the probably subtle structural features that may allow ALBA-independent efficient m6A-interaction. In this context, a YTH-proximal element in the IDRs of mammalian YTHDF proteins is of particular interest for at least two reasons. First, its location relative to the YTH domain is reminiscent of the YAIM described here for plant YTHDFs. Second, it is predicted by AlphaFold (Jumper et al, 2021) to engage in YTH domain interactions, perhaps via disorder-to-order transition upon RNA-binding to stabilize the RNA-bound form (Sikorski et al, 2023), as observed for the *Schizosaccharomyces pombe* YTH domain protein Mmi1 (Wang et al, 2016). This may also be the core function of the YAIM in ECT2, assisted in vivo by ALBA proteins, as deletion of a small YAIM-containing region of ECT2 reduces its m6A-binding capacity in vitro (Seigneurin-Berny et al, 2024). The existence of non-ALBA facilitators of YTHDF-m6A binding in other organisms should not be entirely discarded, however. The mammalian IGF2BP/IMP/ZBP family of RBPs has been suggested to act as m6A readers based on multiple lines of evidence, including m6A-dependent target mRNA association and the similar positions of m6A sites and IGF2BP2 CLIP sites in 3′-UTRs of target mRNAs (Huang et al, 2018). Because the m6A mapping methodology used at the time had limited resolution, it is possible that m6A sites are, in fact, adjacent to IGF2BP2 CLIP sites, particularly since the IGF2BP/IMP/ZBP recognition element (CAUH) defined in previous transcriptome-wide studies (Hafner et al, 2010) is not identical to the DRACH m6A consensus site. The slight off-set between IGF2BP CLIP site and m6A distributions (Huang et al, 2018) is indeed reminiscent of the 3′-UTR distributions of m6A sites and ALBA4 iCLIP sites observed here, and the identification of IGF2BP2 as a prominent interactor of YTHDF1/2/3 in IP-MS experiments (Zaccara and Jaffrey, 2020) is more easily reconciled with a function in facilitated m6A binding by YTHDFs than direct m6A binding competing with YTHDFs. Thus, in light of our results on the ALBA-YTHDF-m6A module in plants, it may be appropriate to consider whether facilitated m6A-reading by YTHDF proteins could have evolved independently in several eukaryotic lineages, and, for mammals in particular, whether a function as a facilitator of m6A reading might explain many of the results originally interpreted to reveal a direct reader function of the IGF2BPs (Huang et al, 2018).

A second important question concerns molecular functions fulfilled by ALBA proteins independently of YTHDF proteins. Such functions are anticipated for a number of reasons. First, while most ALBA4 mRNA binding sites in 3′-UTRs appear to be linked to m6A sites, binding sites in open reading frames were even more numerous and were found in mRNAs with no evidence of m6A modification or ECT2/3 binding. Indeed, ALBA proteins have been found to play a role in heat adaptation via the regulation of Heat Shock Factor-encoding mRNAs, primarily with binding sites in open reading frames (Tong et al, 2022). Second, even the YTHDF-linked ALBA functions may involve properties in addition to assisted m6A-binding, because many ECT2-associated proteins were depleted in the immunoaffinity-purified fraction of the ECT2^YAIM mutant defective in ALBA interaction. Finally, we note that while this report identifies a molecular role of the ALBA domain, it does not address the function of the C-terminal IDR of

long ALBA proteins, expected to be of considerable biological importance given the stronger phenotypes of *alba456* compared to *alba123* mutants, as reported here and by others (Tong et al, 2022). Finally, we note that while this report clearly demonstrates the role of ALBA proteins in the facilitation of m6A reading, it does not exclude the possibility that other RBP families co-purifying with ECTs could also play a role in this process.

# Methods

**Reagents and tools table**

| Reagent/resource | Reference or source | Identifier or catalog number |
| --- | --- | --- |
| **Experimental models** | | |
| *Arabidopsis thaliana* Col-0 | N/A | N/A |
| *E. coli* DH5α cells | Invitrogen | 18265-017 |
| *E. coli* BL21 cells | NEB | C2530H |
| *Agrobacterium tumefaciens* strain GV3101 | (Koncz and Schell, 1986) | N/A |
| **Recombinant DNA** | | |
| pCAMBIA3300-U vector | (Nour-Eldin et al, 2006) | N/A |
| pGEM-T Easy | Promega | A137A |
| pGGD003 | Addgene | 48835 |
| pGGE000 | Addgene | 48860 |
| pGGF003 | Addgene | 48844 |
| pGGZ001 | Addgene | 48868 |
| His6-SUMO pET24 | Addgene | 29711 |
| pDONR/Zeo | Thermo Fisher | 12535035 |
| pMDC111 | (Curtis and Grossniklaus, 2003) | N/A |
| pMDC164 | (Curtis and Grossniklaus, 2003) | N/A |
| **Antibodies** | | |
| anti-ECT2 | (Arribas-Hernández et al, 2018) | N/A |
| anti-ALBA1 | This study | N/A |
| anti-mCherry | Abcam | ab183628 |
| Goat anti-Rabbit IgG | Sigma-Aldrich | A6154-1ML |
| **Oligonucleotides and other sequence-based reagents** | | |
| Primers | This study | Dataset EV6 |
| **Chemicals, enzymes, and other reagents** | | |
| KAPA Hifi Hotstart Uracil + ReadyMix | Roche | KK2801 |
| Phusion High-Fidelity DNA Polymerase | NEB | M0530L |
| BsaI-HF | NEB | R3733S |
| T4 DNA-Ligase | Thermo Scientific | 15224017 |
| KOD Hot Start DNA Polymerase | Sigma-Aldrich | 71086 |
| Gateway BP Clonase II | Thermo Fisher | 11789020 |
| Gateway LR Clonase™ II Plus enzyme | Thermo Fisher | 12538120 |
| MS-agar | PhytoTech Labs | M407 |

| Reagent/resource | Reference or source | Identifier or catalog number |
|---|---|---|
| Tryptone | BD | 211705 |
| Yeast extract | BD | 212750 |
| Agar | BD | 214010 |
| Glufosinate ammonium | Sigma | 45520 |
| Ampicillin | Serva | 13398.02 |
| Kanamycin | Serva | 26898.03 |
| IPTG | Thermo Fisher | R0392 |
| Ni-NTA agarose beads | macherey-nagel | 745400.25 |
| Gravity column | Bio-Rad | 7326008 |
| Imidazole | Thermo Scientific | J17525 |
| His$_6$-tagged ULP1 protease | Kind gift from Prof. Birthe Kragelund, UCPH. | N/A |
| Complete protease inhibitor EDTA-free | Sigma-Aldrich | 11873580001 |
| NuPAGE™ LDS Sample Buffer | Thermo Fisher | NP0007 |
| 4–20% Criterion™ TGX™ Precast gel | Bio-Rad | 5671094 |
| Amersham Protran Premium nitrocellulose membrane | GE Healthcare Life Sciences | GE10600002 |
| TRIzol | Sigma | T9424 - 200 ML |
| RQ1 RNase-Free DNase | Promega | M6101 |
| RNaseOut™ Recombinant RNase Inhibitor | Invitrogen | 10777019 |
| QIAgen RNeasy mini kit | Qiagen | 74104 |
| RNeasy MinElute Cleanup Kit | Qiagen | 74204 |
| SuperScript® III Reverse Transcriptase | Invitrogen | 18080093 |
| RNaseOut™ | Thermo Fisher | 10777019 |
| SensiFAST SYBR No-ROX Mix | Bioline | BIO-98005 |
| RFP-trap agarose beads | Chromotek | rta-20 |
| GFP-trap agarose beads | Chromotek | gta-20 |
| Sepharose 4B beads | Sigma-Aldrich | 4B200 |
| T4-polynucleotide kinase | Thermo Fisher | EK0032 |
| Turbo DNase | Invitrogen | AM2238 |
| Ambion RNase I | Invitrogen | AM2294 |
| T4 RNA ligase 1 high concentration | NEB | M0437M |
| Ribolock | Thermo Fisher | EO0381 |
| γ-$^{32}$P- ATP | Hartmann Analytic | SRP-301-50 |
| 4–12% NuPAGE Bis-Tris gel | Invitrogen | NP0321BOX |
| Proteinase K | Roche | 3115887001 |
| Phenol:chloroform:isoamyl alcohol (25:24:1) | Sigma-Aldrich | P3803-400ML |
| MyONE Silane beads | Thermo Fisher | 37002D |
| ProNex beads | Promega | NG2001 |
| RLT buffer | Qiagen | 79216 |
| RNA Clean and Concentrator-5 kit | Zymo Research | R1015 |
| Ultra low-range GeneRuler | Thermo Fisher | SM1212 |

| Reagent/resource | Reference or source | Identifier or catalog number |
|---|---|---|
| Page Ruler prestained protein ladder | Thermo Fisher | 26616 |
| 0.45 µm filter | Fisher Brand | 15216869 |
| MG-132 | Sigma-Aldrich | M7449 |
| PMSF | Carl Roth | 6367.1 |
| Plant Protease Inhibitor cocktail | Sigma-Aldrich | P9599 |
| DTT | Sigma-Aldrich | D0632-10G |
| SDS | JT Baker | 4095-02 |
| Tris | AppliChem | A1086.1000 |
| HCl | VWR | 20252.290 |
| NaCl | VWR chemicals | 27810.295 |
| MgCl$_2$ | Merck | 1-05833.1000 |
| MOPS | Fisher BioReagents | BP308-500 |
| EDTA | Sigma-Aldrich | 27285-1KG-R |
| Igepal (NP-40) | Sigma | I8896-100 |
| DMSO | Carl Roth | 4720.1 |
| β-Mercaptoethanol | Sigma-Aldrich | M6250 |
| Tween-20 | Sigma | P9416-100ML |
| **Software** | | |
| Rotor-Gene 6000 series software | QIAGEN | |
| ClustalW | (Thompson et al, 1994) | |
| AlphaFold3 | (Abramson et al, 2024) | |
| R | https://www.R-project.org/ | |
| FastQC | https://www.bioinformatics.babraham.ac.uk/projects/fastqc/ | |
| AWK, Mawk 1.3.4 | | |
| Flexbar (3.5.0) | (Roehr et al, 2017) | |
| STAR | (Dobin et al, 2013) | |
| umi_tools (1.0.1) | https://umi-tools.readthedocs.io/en/latest/index.html | |
| samtools (1.14) | (Li et al, 2009) | |
| PureCLIP (1.3.1) | (Krakau et al, 2017) | |
| bedtools (2.27.1) | (Dale et al, 2011; Quinlan and Hall, 2010) | |
| Salmon | (Patro et al, 2017) | |
| hyperTRIBER | (Rennie et al, 2021) | |
| R-package eulerr | (Larsson and Gustafsson 2018; Larsson, 2022) | |
| Tensorflow | (Abadi et al, 2016) | |
| Keras | https://github.com/keras-team/keras | |
| R-package universalmotif | (Tremblay, 2024a; Tremblay, 2024b) | |
| IGV (Integrative Genomics Viewer) | (Robinson et al, 2011) | |
| R-package DESeq2 | (Love et al, 2014) | |
| R-package tidyverse | (Wickham et al, 2019) | |
| R-package GenomicRanges | (Lawrence et al, 2013) | |
| R-package BSgenome | https://bioconductor.org/packages/release/bioc/html/BSgenome.html | |
| R-package INLA | (Rue et al, 2009) https://www.r-inla.org/ | |

| Reagent/resource | Reference or source | Identifier or catalog number |
|---|---|---|
| **Other** | | |
| QIAGEN Rotor-Gene-Q real-time PCR machine | Qiagen | |
| CL-3000 UVP cross-linker | Analytik Jena | |
| Typhoon FLA7000 | GE Healthcare | |
| HiLoad Superdex™ 200 10/300 GL prep grade column | GE Healthcare | |
| HPLC ÄKTA Purifier system | GE Healthcare | |
| NextSeq sequencer | Illumina | |

## Plant material and growth conditions

All lines used in this study are in the *Arabidopsis thaliana* Col-0 ecotype. The following mutant and transgenic lines mentioned have been previously described: *ect2-1 ect3-1 ect4-2* (*te234*) (Arribas-Hernández et al, 2018), *ect2-1 ECT2^{W464A}-mCherry* (Arribas-Hernández et al, 2018), *ect3-2 ECT3-Venus* (Arribas-Hernández et al, 2018), *ect2-1 HA-ECT2* (Tankmar et al, 2023). The *alba1-1*(GABI_560B06), *alba1-2* (SALK_069210), *alba2-1* (GABI_128D08), *alba2-2* (SALKseq_069306), *alba3-1* (SAIL_649_E11), *alba4-1* (SALK_015940), *alba5-1* (SALK_088909), and *alba6-1* (SALK_048337) single mutants were obtained from the Arabidopsis Biological Resource Center (ARBC). Seeds were sterilized by immersing them in 70% EtOH for 2 min, followed by incubation in 1.5% NaOCl, and 0.05% Tween-20 for 10 min, after which the seeds were washed twice with $H_2O$. The seeds were then spread on plates containing Murashige & Skoog (MS) medium (4.1 g/l MS salt, 10 g/l sucrose, 8 g/l Bacto agar). The plates were stratified in darkness at 4 °C for 2–5 days before transfer to Aralab incubators at 21 °C, with a light intensity of 120 µmol/m² and a photoperiod of 16 h light/8 h dark. When needed, after 10 days of growth, seedlings were transferred to soil and kept in Percival incubators under identical settings.

## Generation of ect2-5 ect3-1 ect4-2 by CRISPR-Cas9 genome engineering

For the targeted creation of an in-frame deletion mutant at the endogenous ECT2 locus, we employed the pKIR1.1 CRISPR-Cas9 system (Tsutsui and Higashiyama, 2017). Two plasmids, pKIR1.1-ect2-N8A and pKIR1.1-ect2-N8B, expressing sgRNAs were constructed by ligating oligonucleotides that target *ECT2* into pKIR1.1, as described (Tsutsui and Higashiyama, 2017). The crRNAs were designed to yield a deletion resembling ECT2^{ΔN8} as closely as possible. The plasmids were then transformed into *ect3-1 ect4-2* mutants, and transformants were selected on MS-agar supplemented with 25 µg/mL hygromycin. After transfer to soil, plants with deletions in *ECT2* were identified via PCR using primers spanning the deletion. Progeny from plants with deletions of the expected size, as confirmed by migration in a 1% agarose gel, were plated on MS supplemented with 25 µg/mL hygromycin. Hygromycin-sensitive plants, indicative of the absence of Cas9 and homozygosity of the deletion, were rescued and transferred to MS-agar for recovery. Subsequently, these plants were genotyped and Sanger sequenced for identification of in-frame deletions. Western

blotting, utilizing antibodies raised against synthetic peptides in the ECT2 IDR outside the deleted region (Arribas-Hernández et al, 2018), was performed to confirm the in-frame deletion. Primers are listed in Dataset EV6.

## Construction of transgenic lines

To generate the constructs *pro(ALBA2):ALBA2-FLAG-TFP:ter(-ALBA2)*, *pro(ALBA4):ALBA4-VENUS:ter(ALBA4)*, *pro(ALBA2):ALBA2-FLAG-ADAR:ter(ALBA2)*, *pro(ALBA4):ALBA4-FLAG-ADAR:ter(ALBA4)*, *pro(ECT2):ECT2^{YAIM}-mCherry:ter(ECT2)*, PCR-amplified DNA fragments were pieced together by USER cloning (Bitinaite and Nichols, 2009) in all cases except for *pro(ECT2):ECT2^{YAIM}-mCherry:-ter(ECT2)* in which an appropriate dsDNA containing the YAIM-mutations was synthesized (Integrated DNA Technologies, gBlocks). As a template for PCR, we used plasmids containing wild-type *pro(ECT2):ECT2-mCherry:ter(ECT2)* (Arribas-Hernández et al, 2018) for *ECT2-mCherry* constructs, *pro(ECT2):ECT2-FLAG-ADAR:ter(-ECT2)* for *FLAG-ADAR* constructs (Arribas-Hernández et al, 2021a), and *pro(ECT3):ECT3-VENUS:ter(ECT3)* for *VENUS* constructs (Arribas-Hernández et al, 2018). DNA fragments were amplified using dU-substituted primers and KAPA Hifi Hotstart Uracil+ ReadyMix (Bitinaite and Nichols, 2009). The amplified fragments were inserted into the pCAMBIA3300-U vector, a modified version with a double PacI USER cassette (Nour-Eldin et al, 2006). To clone *pro(AL-BA1):ALBA1-FLAG-TFP:ter(ALBA1)*, we made use of Greengate cloning. Briefly, PCR fragments were amplified using Thermo Scientific Phusion High-Fidelity DNA Polymerase (NEB) and ligated into entry vectors through BsaI-restriction cloning. The *pro(AL-BA1):ALBA1* gDNA fragment was subcloned into pGEM-T Easy by A-tailing (Promega) prior to BsaI-restriction cloning. The vectors containing *pro(ALBA1):ALBA1* (in pGEM-T Easy), linker-*TFP* (pGGD003), *ALBA1* 3′UTR and downstream sequences (in pGGE000), and the D-AlaR cassette (pGGF003) were combined in a 'Greengate reaction' using BsaI-HF (NEB), T4 DNA-Ligase (Thermo Scientific), and pGGZ001 as the destination vector. *pro(AL-BA2):ALBA2-FLAG-TFP:ter(ALBA2)*, and *pro(ALBA4):ALBA4-FLAG-Venus:ter(ALBA4)* fusions were constructed by USER cloning with the primers listed in Dataset EV6. To clone *ALBA1-GFP, ALBA2-GFP, ALBA4-GFP*, and *ALBA5-GFP* used for confocal microscopy, we employed Gateway cloning. *ALBA* gene fragments, including 5′-regions, exons/introns to the gene's end (excluding the stop codon), were amplified with attB1 and attB2 sites for Gateway cloning using KOD Hot Start DNA Polymerase. Purified amplicons were cloned into pDONR/Zeo via Gateway BP Clonase II (Thermo Fisher) and transformed into *E. coli* α-select cells. Subsequently, entry clones were recombined with the destination vectors pMDC111 and pMDC164, respectively (Curtis and Grossniklaus, 2003; Earley et al, 2006) via Gateway LR Clonase II (Thermo Fisher) to generate expression clones. All plasmids were verified through restriction digestion and sequencing before being transformed into respective plants using Agrobacterium-mediated floral dip (Clough and Bent, 1998). Primers are listed in Dataset EV6.

## Screening for te234 complementation

Screening of primary transformants (T1s) expressing wild-type, deletion, or point mutant variants of ECT2-mCherry in the *te234* background was done as previously described (Tankmar et al,

2023). In brief, primary transformants were selected on MS-agar plates containing glufosinate ammonium (7.5 mg/L (Sigma)) to select plants with the transgene and ampicillin (10 mg/l) to restrict agrobacterial growth. Nine days after germination, primary transformants were categorized according to the size(s) of the first true leaves: full complementation ($s \geq 1$ mm), partial complementation (0.5 mm $< s <$ 1 mm), or no complementation ($s \leq 0.5$ mm). The complementation percentages were then determined by dividing the number of seedlings in each complementation category by the total number of transformants.

## Statistical analysis of complementation data

Statistical significance of the different T1 complementation categories was determined using Fisher's exact test, and the Holm–Bonferroni method was applied to address multiple testing. Student's *t*-test was used to evaluate the significance of differences in leaf size between Col-0 WT, *de34* (*ect3-1 ect4-2*), *te234* (*ect2-3 ect3-1 ect4-2*), and the CRISPR-generated *ect2-5 ect3-1 ect4-2*.

## Analysis of trichome phenotypes

Counts of trichomes with different numbers of branches and the statistical analysis of the raw data were done as described (Arribas-Hernández et al, 2018).

## Western blotting

Western blotting was performed as described (Tankmar et al, 2023). Briefly, 100–300 mg of tissue were ground in liquid nitrogen and resuspended in 5 volumes of IP buffer (50 mM Tris-HCl pH 7.5, 150 mM NaCl, 10% glycerol, 5 mM $MgCl_2$, and 0.1% Nonidet P40), supplemented with 1x protease inhibitor (Roche Complete tablets) and 1 mM DTT. The lysate was centrifuged at 13,000×*g* for 10 min and 4× LDS sample buffer (277.8 mM Tris-HCl pH 6.8, 44.4% (v/v) glycerol, 4.4% LDS, and 0.02% bromophenol blue) was added to a final concentration of 1× LDS. Subsequently, the samples were denatured at 75 °C for 10 min and run on a 4–20% Criterion™ TGX™ Precast gel in 1× Tris-glycine, 0.1% SDS buffer at 90–120 V for ~1 h on ice. The proteins were transferred onto an Amersham Protran Premium nitrocellulose membrane (GE Healthcare Life Sciences) in a cold transfer buffer (1 × Tris-glycine, 20% EtOH) at 80 V for 1 h on ice. The membrane was then blocked in 5% skim milk in PBS-T (137 mM NaCl, 2.7 mM KCl, 10 mM $Na_2HPO_4$, 1.8 mM $KH_2PO_4$, pH 7.4, 0.05% Tween-20) for 30 min. After blocking, membranes were probed with antibodies specific for ECT2 (Arribas-Hernández et al, 2018), ALBA1 (1:2000, see below), ALBA4 (1:1000, see below), or commercially available antibodies against mCherry (Abcam ab183628, 1:2000 dilution) at 4 °C overnight. Membranes were then washed three times in PBS-T, incubated with HRP-coupled goat anti-rabbit antibody and developed using chemiluminescence detection, as previously described (Arribas-Hernández et al, 2018).

## RNA extraction and qRT-PCR

Total RNA was extracted from frozen and ground plant powder using TRIzol® (1 mL per 500 mg sample). 14 µg RNA was treated with 14 µL of RQ1 RNase-Free DNase (Promega) and 1 µL of RNaseOut™ Recombinant RNase Inhibitor (Invitrogen) following

the manufacturer's protocol. The RNA was then purified using the QIAgen RNeasy mini kit following the RNeasy column clean-up protocol. The RNA quantity and quality were determined via NanoDrop and agarose gel electrophoresis. cDNA was prepared using SuperScript® III Reverse Transcriptase (Invitrogen), with the addition of RNaseOut™. For qRT-PCR, 0.4 µL 10 µM specific primer pairs (mixture of forward and reverse primers) was mixed with 10 µL SensiFAST SYBR (Bioline) mastermix and 9.6 µL of cDNA. All the qRT-PCR reactions were performed in three technical replicates, carried out on a QIAGEN Rotor-Gene-Q real-time PCR machine and analyzed with the Rotor-Gene 6000 series software (QIAGEN). *CYCLOPHILIN* (At2g29960) was used for normalization. Primers are listed in Dataset EV6.

## CLIP-PNK assays of ECT2-mCherry variants

Twelve-day-old seedlings were UV-crosslinked with 2000 mJ/cm² and ground into a fine powder in liquid nitrogen. Immunoprecipitation with RFP-trap beads (Chromotek), washes, DNase and RNase digestion, PNK-labeling, SDS-PAGE, membrane transfer, and autoradiography were performed as described in (Arribas-Hernández et al, 2021a). We used 20 µL of beads for 1 g of tissue in 1.5 mL of iCLIP buffer for every sample.

## Immunoprecipitation and LC-MS

Immunoprecipitations of ECT1-TFP, ECT2-mCherry variants, and ECT3-Venus were performed as described by (Tankmar et al, 2023), while immunoprecipitations of ALBA4-GFP or GFP were performed as described by (Speth et al, 2014). Briefly, 7-day-old seedlings expressing ALBA4-GFP or GFP alone were harvested and ground into fine powder using liquid nitrogen. For each replicated, 0.5 g of ground plant tissue was homogenized in 1.5 mL IP buffer (50 mM Tris-HCl pH 7.5, 150 mM NaCl, 10% glycerol, 0.1% Triton-X100) supplemented with 2% (w/v) PVP40, Roche Complete Protease Inhibitor cocktail (1 tablet/50 mL), 100 µM MG-132, 1 mM PMSF and Sigma Plant Protease Inhibitor cocktail (1/30 v/v). Samples were centrifuged at 16,000 × *g* for 5 min at 4 °C, the supernatant was transferred to a new tube, and centrifugation was repeated for 10 min. The supernatant was again transferred to a new tube and filtered through a 0.45-µm filter. For Co-IP, 1 mL of cell extract at a concentration of 2 µg/µL was first added to 50 µL of sepharose beads for pre-clearing and incubated for 30 min at 4 °C with constant rotation. After centrifugation at 1000 × *g* for 1.5 min at 4 °C, the cell extract was added to 20 µL GFP-Trap beads and incubated for 2.5 h at 4 °C with constant rotation. The beads were washed 4x in Co-IP wash buffer (50 mM Tris-HCl pH 7.5, 150 mM NaCl, 10% glycerol, 0.05% Triton-X100, Roche Complete Protease Inhibitor cocktail (1 tablet/50 mL)) and proteins were eluted by addition of 40 µL 2x LDS sample buffer to the beads and incubation at 70 °C for 10 min. For control samples treated with nucleases, beads were washed once in Co-IP wash buffer (+10 mM $MgCl_2$) after the IP. Beads were then resuspended in 100 µL Co-IP wash buffer (+10 mM $MgCl_2$) and treated with 2 µL Turbo DNase (Thermo Fisher Scientific) and, optionally, 5 µL of a 1:50 dilution of RNase I (Ambion) for 10 min at 37 °C and 1200 rpm. Beads were then washed three times with Co-IP wash buffer, and elution was performed as described above. Mass spectrometry data was analysed as in (Tankmar et al, 2023).

## Protein expression of ALBA1

An ALBA1 (AT1g29250) cDNA was amplified from oligo(dT)-primed reverse transcription products of DNase-treated total RNA from Col-0 wild type using the primer set MT303-MT304. The resulting PCR product was ligated in a frame downstream of His$_6$-SUMO in a pET24-derived vector containing His$_6$-SUMO (Twist Bioscience). For recombinant protein expression, the plasmid encoding His-SUMO-ALBA1 was transformed into *E. coli* BL21 (DE3 7tRNA) codon plus. Cells were grown at 37 °C in LB medium supplemented with 35 µg/ml kanamycin, and expression was induced at $OD_{600} \approx 0.6$ by the addition of 0.5 mM IPTG. Following induction, the cells were grown at 18 °C overnight and harvested by centrifugation. The cell pellet was resuspended in 20 mM Tris-HCl (pH 8), 10 mM imidazole, and 300 mM NaCl supplemented with 1 mM DTT and EDTA-free protease inhibitor (cOmplete; Roche). Cells were lysed once using a French press (20,000 psi). Crude lysate was cleared by centrifugation at $30,000 \times g$ for 30 min at 4 °C and filtered through a 0.45-µm membrane. His-SUMO-ALBA1 was purified on $Ni^{2+}$-NTA resin by incubation for 1 h at 4 °C, after which the beads were washed in wash buffer (20 mM Tris-HCl pH 8, 20 mM imidazole, 200 mM NaCl), and the bound protein was eluted in elution buffer (300 mM imidazole, 20 mM Tris-HCl pH 8, 300 mM NaCl). The eluted protein was dialyzed overnight into 20 mM Tris-HCl pH 8, 200 mM NaCl, 1 mM 2-mercaptoethanol followed by cleavage after the His$_6$-SUMO tag with heterologously expressed His$_6$-tagged ULP1 protease, a kind gift from Birthe Kragelund. $Ni^{2+}$-NTA resin was used to bind the protease and impurities bound to the $Ni^{2+}$-NTA resin in the first affinity purification, and ALBA1 was collected in the flowthrough. ALBA1 was further purified on a HiLoad Superdex™ 200 10/300 GL prep grade column (GE Healthcare) connected to an HPLC ÄKTA Purifier system (GE Healthcare). Eluates were monitored at $A_{280}$, and purity was assessed by SDS-page analysis.

## Development of ALBA1 and ALBA4 antibodies

The anti-ALBA1 and anti-ALBA4 antibodies were affinity-purified by Eurogentec from serum collected from rabbits immunized with recombinant ALBA1 protein or a 1:1 mix of the KLH-coupled ALBA4 peptides H-CGFNNRSDGPPVQAAA-OH and H-CNGPP NEYDAPQDGGY-NH$_2$ (Eurogentech). The ALBA4 peptides were synthesized by Schafer-N Aps, Copenhagen, Denmark.

## Protein alignment and logo representation

All YTHDF protein sequences from 36 plant species, or selected DF-A, -B, -C, -D, and -E protein sequences from 34 plant species (Dataset EV7) were aligned using MUSCLE (Madeira et al, 2024). The resulting alignments were trimmed from both ends using Jalview (Waterhouse et al, 2009), to leave the region comprising the ECT2 N8 motif. Short and under-represented amino acid insertions not contained in ECT2 were also removed for clarity (positions marked as blue arrowheads in Appendix Fig. S3). To build the N8 motif logo, we used the online tool Weblogo (Crooks et al, 2004).

## Structural modeling using AlphaFold3

The structural model of the ECT2-(ALBA5)$_2$-RNA complex was generated by AlphaFold3 (Abramson et al, 2024) using default settings and the following sequence input: one molecule of ECT2 (gene model AT3g13460.1, amino acid residues 373–616), two molecules of ALBA5 (gene model AT1g20220.1, amino acid residues 18–114), one molecule of RNA (5′-AAA[m$^6$A]CUU-CUG-3′).

## ALBA4-GFP iCLIP experiments and library preparation

iCLIP experiments were carried out based on the method previously employed for Arabidopsis GRP7-GFP (Meyer et al, 2017) and the optimized iCLIP2 protocol (Buchbender et al, 2020; Lewinski et al, 2024). Briefly, 7-day-old seedlings expressing ALBA4-GFP or GFP alone grown at 20 °C in LD (16 h light, 8 h dark) were crosslinked with 254 nm UV light at 2000 mJ/cm², snap frozen, and ground into a fine powder in liquid nitrogen, and homogenized in iCLIP lysis buffer (50 mM Tris-HCl pH 7.5, 150 mM NaCl, 4 mM MgCl$_2$, 5 mM DTT, 1% SDS, 0.25% sodium deoxycholate, 0.25% Igepal) supplemented with Roche Complete Protease Inhibitor cocktail (1 tablet/50 mL). The lysate was cleared by centrifugation and filtration (0.45-µm pore) of the supernatant. After pre-clearing with 200 µL of sepharose beads for 1 h at 4 °C, RNP complexes were immunopurified with GFP-Trap beads (ChromoTek) for 4 h at 4 °C under constant rotation. We used 50 µL of beads for 3 g of tissue in 5 mL of iCLIP lysis buffer for every replicate. After washing four times with iCLIP wash buffer (2 M urea, 50 mM Tris-HCl pH 7.5, 500 mM NaCl, 4 mM MgCl$_2$, 2 mM DTT, 1% SDS, 0.5% sodium deoxycholate, 0.5% Igepal, supplemented with Roche Complete Protease Inhibitor cocktail (1 tablet/50 mL)), and twice with PNK wash buffer (20 mM Tris-HCl, pH 7.4, 10 mM MgCl$_2$, 0.2% Tween-20), RNP complexes attached to the beads were subjected to treatment with DNase (Turbo DNase [Ambion], 4 U/100 µL) and optionally RNase I (Ambion, 1 U/mL) at 37 °C for 10 min. Subsequently, RNA 3′-ends were dephosphorylated (PNK [Thermo Fisher] in buffer containing 350 mM Tris-HCl pH 6.5, 50 mM MgCl$_2$, 25 mM DTT) for 20 min at 37 °C, followed by one wash with PNK wash buffer, one wash with high-salt buffer (50 mM Tris-HCl pH 7.4, 1 M NaCl, 1 mM EDTA, 1% Igepal, 0.1% SDS, 0.5% sodium deoxycholate) and two more washes with PNK wash buffer. The L3 linker was then ligated to the 3′-RNA ends (with NEB HC RNA Ligase in ligation buffer (200 mM Tris-HCl pH 7.8, 40 mM MgCl$_2$, 40 mM DTT with RiboLock and PEG8000) at 16 °C and 1250 rpm for >16 h.

Samples were then washed twice in high-salt buffer and once in PNK wash buffer before the RNA was radioactively labeled at the 5′-end by PNK-mediated phosphorylation using γ-$^{32}$P- ATP (20 min at 37 °C). The labeled RNP complexes were subjected to SDS-PAGE (4–12% NuPAGE Bis-Tris gel with 1x MOPS buffer) and blotting on a nitrocellulose membrane (Protran BA-85). Pieces of membrane containing a size range of RNA species bound to the protein (a smear above the expected molecular weight localized by autoradiography) were excised and subjected to proteolysis (200 µg of Proteinase K [Roche] in 200 µL of PK buffer [100 mM Tris-HCl pH 7.4, 50 mM NaCl, 10 mM EDTA] for 20 min at 37 °C) to release RNA bound to small peptides. The RNA was then purified using phenol-chloroform (pH 7.0) and ethanol precipitation and used to prepare sequencing libraries following the iCLIP2 protocol (Buchbender et al, 2020): reverse transcription with SSIII (Invitrogen) and an RT oligo complementary to the L3 liker followed by RNA hydrolysis and cDNA clean-up with MyONE Silane beads (Thermo Fisher). A second

adapter was then ligated to the 3'OH of the cDNAs (with NEB HC RNA Ligase in NEB ligation buffer plus 5% DMSO, 1 mM ATP, and 22.5% PEG8000) at 20 °C and 1250 rpm overnight. The adapter contains a bipartite unique molecular identifier (UMI) and an experimental barcode, allowing for PCR duplicate removal and sample multiplexing, respectively. After another MyONE Silane clean-up, the cDNA library was pre-amplified in a first PCR (6 cycles), followed by size selection with ProNex beads (Promega) to remove short cDNAs and primer dimers. The cDNA library was then amplified in a second PCR, followed by a second ProNex size selection to remove PCR primers and finally prepare the cDNA library for sequencing. The second PCR was carried out with 10 µL of cDNA and eight cycles for each replicate. Samples were multiplexed and sequenced in the NextSeq sequencer (NextSeq® 500/550 Mid Output Kit v2 (150 cycles)) at the Genomics Core Facility at IMB (Mainz, Germany).

## ALBA4-GFP iCLIP analysis

All reads from iCLIP experiments were quality-checked after multiple processing steps with FastQC (0.11.9). The distribution of read counts assigned to sample barcodes was computed using awk (GNU awk 5.0.1). Reads were demultiplexed, sequencing adapters removed from 3′ ends and subsequently quality- as well as length-trimmed (--min-read-length 15 -q WIN -qf sanger –min-read-length 15) with Flexbar (3.5.0) while keeping the random UMI parts in the read id field (--umi-tags). A genome index was created using STAR (2.7.3a) using the *Arabidopsis thaliana* genome version TAIR10. The genome annotation from Araport (version 11) was specified to mark the location of splice junctions. Quality trimmed reads were then mapped using STAR and the created genome index, allowing only softclipping of 3′ ends (--alignEndsType Extend5pOfRead1) to preserve the position of the crosslinked nucleotide. PCR duplicates were removed using umi_tools (1.0.1) by considering the UMI tag in the read id field and the mapping coordinates. The uniquely mapped and deduplicated reads from each ALBA4-GFP and GFP replicate were merged together using samtools (1.14) and peak called with PureCLIP (1.3.1) in standard mode (-bc 0) to identify short and defined peak coordinates. In order to learn the HMM parameters, only the first two chromosomes were specified (-iv 'Chr1;Chr2'), and the precision to store probabilities was set to long double (-ld). Clusters of directly adjacent called peaks were merged and reduced to the position with the highest reported PureCLIP score (1-nt resolution). Binding sites were defined as called peaks, extended by 4 nt (−4…0… + 4) in both directions with bedtools (2.27.1). Sites which reported crosslinks in only 1 out of 9 position were removed as they are considered artifacts. To confirm that the binding sites are supported by at least 2 replicates and a sufficient number of reads (reproducible binding sites), the coordinates of binding sites were overlapped with crosslink positions from every replicate (ALBA4-GFP and GFP independently). The distribution of crosslinks per binding sites was used to determine a reproducibility threshold. After defining a distribution quantile of 30% as the minimal filtering threshold, only binding sites above this threshold in at least two out of three replicates were kept. Due to the low amount of uniquely mapped reads the GFP control was not tested for reproducibility. Reproducible binding sites of ALBA4-GFP overlapping with binding sites from the GFP control were removed using bedtools and reported in browser extensible data (BED) format. Targets of ALBA4-GFP were defined as transcripts overlapping reproducible binding sites. Only the locations of representative gene models from Araport (version 11) were considered. For visual

inspection data tracks were generated from uniquely mapped ALBA4-GFP and GFP-only reads using bedtools.

## Sample preparation for TRIBE and HyperTRIBE

RNA extraction and library preparation was performed as previously described(Arribas-Hernández et al, 2021a). Total RNA was extracted from manually dissected root tips for ALBA4-FLAG-ADAR and apices (removing cotyledons) for ALBA2-FLAG-ADAR and ECT2-FLAG-ADAR of five independent lines (10-day-old T2 seedlings) with each of the lines being used as biological replicate.

## TRIBE/HyperTRIBE analyses for ALBA2 and ALBA4 vs. free ADAR controls

For all TRIBE/HyperTRIBE experiments, reads were mapped to the TAIR10 genome using STAR (Dobin et al, 2013) (version 2.7.11) and transcripts quantified using Salmon (Patro et al, 2017) based on the Araport11 transcriptome (Cheng et al, 2017) augmented with the DNA sequence for the ADAR clone. The hyperTRIBER pipeline (Rennie et al, 2021) was employed in order to quantify all positions with at least one mismatch to the genome, filter candidate positions by mutation type (A-to-G or T-to-C for forward or reverse strands, respectively), and replicate agreement, and formally test these candidates using a generalized linear model-based approach for assessing the difference in editing proportions between free ADAR control samples vs. fusion samples, retaining positions with a $\log_2$FC >1, an adjusted $p$ value <0.01 and a minimum editing proportion of 0.01. All sets were further annotated using the hyperTRIBER pipeline based on Araport11 gene annotations and prioritizing highly expressed transcripts in the control lines in the case of positions overlapping multiple transcripts.

## HyperTRIBE analysis for ECT2 in the *alba1245* background and ALBA2 in the *Gte234* background

Unequal levels of *ECT2-FLAG-ADAR* or *ALBA2-FLAG-ADAR* expression between different genetic backgrounds in the same HyperTRIBE experiment could result in misinterpretation of results due to biased ADAR-driven editing patterns. This was supported by inspection of the initial results from the hyperTRI-BER pipeline (Rennie et al, 2021) when comparing *ECT2-FLAG-ADAR*-expressing plants in the Col-0 vs *alba1245* backgrounds. This preliminary analysis showed stronger editing in the direction of the samples with higher average *ADAR* expression, supported by western blots. To investigate further, we first re-ran the pipeline on only four lines (two per genetic background), selected such that the average number of reads mapping to *ADAR* was approximately equal between the two genetic backgrounds. Compared to the naïve analysis of all five lines per genotype, the significantly differently edited sites were visually less biased in the direction of high *ADAR* expression, indicating that unequal *ADAR* expression leads to spurious results if left uncorrected. Furthermore, we observed a pattern whereby sites on lowly expressed genes tended to exhibit a larger editing proportion. To robustly account for differences in *ADAR* expression as measured by mRNA-seq read counts, we formulated a Bayesian hierarchical model as follows. First, we split the samples into three groups according to the expression of the ADAR clone (ADAR_BIN) and binned expression levels into five

groups (EXPR_BIN). Let $Y_{ijkc}$ denote the observed count of base $G$ at the $i$-th position, with the $j$-th level of ADAR_BIN, the $k$-th level of EXPR_BIN, and under condition $c$. $Y_{ijkc}$ is assumed to follow a Binomial distribution $Y_{ijkc} \sim \text{Binomial}(n_{ijkc}, p_{ijkc})$ where $n_{ijkc}$ represents the number of trials for each combination of position, ADAR_BIN level, EXPR_BIN level, and condition, and $p_{ijkc}$ is the probability of observing base $G$. Then the logit of $p_{ijkc}$ is modeled as $\log\left(\frac{p_{ijkc}}{1-p_{ijkc}}\right) = \eta_{ijkc}$ where the linear predictor $\eta_{ijkc}$ is given by:

$$\eta_{ijkc} = \beta_0 + \beta_j + \gamma_k + \delta_{jk} + u_{ic}$$

where $\beta_0$ is the intercept, $\beta_j$ is the effect of ADAR bin $j$, $\gamma_k$ is the effect of expression bin $k$, $\delta_{jk}$ is the corresponding ADAR expression interaction and $u_{ic} \sim N(0, \tau_c^{-1})$ is a position-specific random effect with condition-specific precision parameter $\tau_c$. The model was fit using the Integrated Nested Latent Laplace (INLA) framework.

Let $u_{iA}$ and $u_{iB}$ denote the random effects for position $i$ under conditions A and B, then the linear combination is $LC_i = u_{iA} - u_{iB}$ was computed from the posterior distribution of the fitted model. The mean $\mu_{LC_i}$ and standard deviation $\sigma_{LC_i}$ of samplings from the fitted posterior were used to generate Z-scores $Z_i = \frac{\mu_{LC_i}}{\sigma_{LC_i}}$ which were converted into $p$ values and subsequently adjusted to a false discovery rate. Importantly, the list of significant genes from this analysis strongly overlapped with the smaller list of genes from the two-sample analysis described above (Fig. EV4F,G).

Finally, position-specific corrected editing proportions from the fitted model were further estimated by assuming ADAR to be exactly to the center bin and used for producing scatter plots for all tested positions.

### Definitions of strict and permissive gene sets

Strict sets: ALBA4, the intersection of iCLIP (strong) and ALBA4-TRIBE-associated gene sets. ECT2, intersection between ECT2/3 HyperTRIBE and ECT2 iCLIP (110 KDa) target sets (Arribas-Hernández et al, 2021b). Permissive sets: union instead of intersection between the above sets for ALBA4 and ECT2, respectively.

### Definition of matched background sets

For sites of interest (iCLIP crosslink sites, m⁶A site positions, etc.), enrichment of surrounding features (motifs, sites of interest) was assessed by comparing the same features around matched background sets. These sets were calculated similarly to our previous publication (Arribas-Hernández et al, 2021a), where, briefly, each site of interest was paired with a site of similar proportion along the relevant transcript feature (5′-UTR, CDS, or 3′-UTR). To avoid expression bias, the pool of background genes was limited to those of the set of interest, or alternatively based on a set of genes selected to have a similar expression distribution.

### Venn diagrams and significance of overlaps

Venn diagrams were generated using custom code and the R-package eulerr (https://CRAN.R-project.org/package=eulerr) (Larsson and Gustafsson 2018; Larsson, 2022). To assess the significance of overlaps between two sets of genes, a random set of genes of size equal to the number of genes in the first set was selected. To avoid expression bias—due to random genes being, on average, more lowly expressed than the sets of interest—the expression distribution of the random set was matched to that of the first set. We calculated the number of genes in the first set overlapping with the second set, as well as the number of genes for each of the 1000 random samples overlapping with the second set.

The $p$ value was calculated as: $p - \text{value} = 1 - 2 \times \left(\frac{\left|hsum - \frac{1000}{2}\right|}{1000}\right)$

where $hsum$ is the number of cases where the number of genes in the random set overlapped more with the second set. In cases where there were zero instances where the random set had a better overlap with the second set, the $p$ value was set to "<0.001", indicating a high significance of overlap. This procedure was carried out using a custom script, which also returned a single random set of expression distribution matched genes. This random set was used in the Venn diagrams to provide a visual indication of the expected overlap by chance.

To check for possible false positives in genes with fewer than 2–5 iCLIP sites, we overlapped the set with the ALBA4 HyperTRIBE data and looked for the percentage of support. We noted that genes with only a single, low-quality iCLIP site tended to be supported by ALBA4 HyperTRIBE to a similar level as random sets of expressed genes, providing justification for considering the more robust set for subsequent analyses.

### Metagene plots

Metagene plots showing enrichment of features in 5′-UTR, CDS and 3′-UTR corrected for the size of the annotated region were generated using a strategy similar to what we previously reported (Arribas-Hernández et al, 2021a).

### Single-cell co-expression analysis of ECT2

We first obtained single-cell mRNA-seq root tip data (He et al, 2023; Shahan et al, 2022). To avoid bias due to differences in UMI count between ECT2-expressing (ECT2+) cells and non-expressing (ECT2-) cells, each ECT2+ cell was matched with an ECT2- cell of similar UMI count. For each gene G expressed within the range of 20–80% of the resulting total cells, counts of G+ and G− cells for each of the ECT2+ and ECT2- sets were used to perform a Fisher's exact test, whereby a high odds ratio represents a high corresponding between ECT2 and the tested G, indicative of co-expression.

### Motif analysis

We first considered the set of motifs previously defined on the basis of ECT2 iCLIP data (Arribas-Hernández et al, 2021a). Background sites for m⁶A (nanopore-derived (Parker et al, 2020)), ALBA4 iCLIP, and ECT2 iCLIP were generated following a similar strategy to what we previously reported (Arribas-Hernández et al, 2021a), ensuring that the distribution of site locations across gene features were identical for both the true set and the background set. We subsequently removed background sites which, by chance, overlapped with sites from the true sets (within 100 bp). For both the true sets and background sites, we calculated the number of motifs

present per 1000 sites (a normalization allowing for comparability across different sets), for each position up to 100 bp from the site.

## Curation of m⁶A site set

We first collected m⁶A sites for *A. thaliana* from multiple published sources (Parker et al, 2020; Tang et al, 2020). As nanopore-derived sites are not subjected to UV-bias, we trained a neural network to differentiate between the 20,858 m⁶A sites identified by nanopore (Parker et al, 2020) and a corresponding set of 20,715 location-matched negative sites (Fig. EV3A). The neural network used as input extracted sequence ±100 bp regions around all positions (R-packages BSgenome (Pagès, 2024) and AThaliana), which was converted from FASTA to one-hot encoded format (Pedregosa et al, 2011). As output, the network predicted the presence or absence of the m6A at the center point of the input sequence. The network was based on four blocks of 1D convolutional layer with ReLu activation, batch normalization, and max pooling of size 2. The output of these four blocks was flattened, run through a fully connected layer, and then passed into a fully connected output layer of output size 1 with a sigmoid activation function. The model was trained specifying the binary cross entropy loss function using Keras with a Tensorflow backend (Abadi et al, 2016) specifying binary cross entropy loss function. The model showed excellent performance, with AUC ranging from 0.85 to 0.92 over the five folds. This model enabled us to fine-adjust sites from other sets by systematically shifting their positions and selecting those with the highest probability (Fig. EV3B). Consequently, we augmented the smaller set of nanopore-derived positions with a broader set exhibiting properties highly consistent with nanopore-identified sites (Fig. EV3C–F). Notably, ~90,000 miCLIP-derived positions not only shifted to locations similar to nearby nanopore-defined sites, but also consolidated into fewer positions, indicating that many miCLIP-identified sites represent imprecise locations. Overall, our augmentation strategy yielded a compendium of 41,883 m⁶A sites in *A. thaliana*.

## Convolutional neural network-based de novo motif detection

We annotated each of 41,883 m⁶A sites as bound or unbound according to an overlap of either ECT2 iCLIP or ALBA iCLIP sites within 100 nt. For each task (task 1: predicting the presence of ECT2 binding, task 2: predicting the presence of ALBA4 binding), we defined the positives as m⁶A sites bound by the relevant protein and the negatives as unbound m⁶A sites. For each m⁶A site, 300 nt of sequence was extracted on either side, creating 601 nt long sequences, which were embedded using one-hot encoding and passed as input in a convolutional neural network with two outputs —(1) presence or absence of ECT2 iCLIP and (2) presence or absence of ALBA iCLIP. The network architecture consisted of five blocks of 1D convolutional layers with RELU activation, a 0.2 drop-out layer, a batch normalization layer, and a max-pooling layer with pool size 2. Each convolutional layer had 64 filters, with a kernel size of 8 in the first layer and 6 thereafter. The output was then flattened into one dimension and passed through a separate connected layer of kernel 32 for each output, which was specified as a fully connected layer of size 1 using a sigmoid activation function. The network was trained using Keras with a Tensorflow backend

(Abadi et al, 2016), specifying the binary cross entropy loss function for each output.

## Fivefold cross-validation strategy for machine learning models

Sites were split into five sets of similar size. Since there are often multiple m⁶A sites on a single gene, and these sites often fall within overlapping windows, we separated training and test sets such that no gene was present in both sets. This has the effect of ensuring that sequences were fully separated between training and testing sets, so that performance on the testing sets could be fairly evaluated without being inflated by possible overtraining. Each testing set consisted of one of the five sets, and the training set the remaining sets combined. All predictions used in subsequent analyses were based only on sets held out of the training process.

## Modeling of RBP-specific motifs

After fitting each fold weights for the 32 learned convolutional filters of length 8 from the initial layer (that is, the layer connecting to the one-hot encoded input sequence) were extracted, resulting in a total of 160 filters. For each of these filters individually, we scanned through all sequences from the training set, selected the top 5000 high-scoring positions, and used the resulting nucleotide frequencies at each of the 8 positions to derive a position weight matrix. These position weight matrices were then allocated a consensus name using the R-package universalmotif (Tremblay, 2024).

For each motif, m⁶A-centered sequences were classified as containing or not containing the given motif within 150 bp of the methylation site. To detect RBP-specific binding motifs, a generalized linear model (glm) assuming a binomial-distributed response (logistic regression) was used to predict motif presence as the dependent variable, where the two predictors in the model were the probability of ECT2 binding from the neural network and the probability of ALBA4 binding from the neural network. In this way, the coefficient for ECT2 binding is interpreted as the strength of correspondence with that motif whilst controlling for binding of ALBA4, and vice-versa. Z-scores for each of the two proteins for all motifs were then extracted from the model and plotted as enrichment scores.

# Data availability

All sequencing data have been deposited in the European Nucleotide Archive under accession code PRJEB71752 (https://www.ebi.ac.uk/ena/browser/view/PRJEB71752). The mass spectrometry proteomics data have been deposited to the ProteomeXchange Consortium via the PRIDE (Perez-Riverol et al, 2021) partner repository with the dataset identifier PXD052232 (http://www.ebi.ac.uk/pride/archive/projects/PXD052232). The code used for data analysis is available at Github: https://github.com/sarah-ku/ALBA_YTH_arabidopsis. The neural network data, including all models, training and testing data sets (in FASTA format), coordinates of m⁶A sites, and predictions are available on Zenodo (https://doi.org/10.5281/zenodo.11241987).

The source data of this paper are collected in the following database record: biostudies:S-SCDT-10_1038-S44318-024-00312-0.

## Peer review information

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

## Acknowledgements

We thank Kristina Neudorf, Lena Bjørn Johansson, Daniel Tobias Kyndesen Lahti, Ida Thorøe Michler, Magnus von Holstein-Rathlou, and Jakub Najbar for their valuable technical assistance and Theo Bölsterli and René Hvidberg Petersen and their teams for plant care. Christian Poulsen is thanked for help with AlphaFold3 modeling. We acknowledge the proteomics platform at Denmark's Technical University for its expertise in running protein identification through liquid chromatography-mass spectrometry. We also thank Mandy Rettel and Frank Stein from the EMBL Proteomics Core facility for IP-MS analysis of ALBA4 IPs. Carlotta Porcelli is thanked for advice on the analysis of mass spectrometry data. Support by the IMB Genomics Core Facility and the use of its NextSeq500 (funded by the Deutsche Forschungsgemeinschaft (DFG, German Research Foundation)—INST 247/870-1 FUGG) is gratefully acknowledged. This research was supported by a Hallas-Møller Ascending Investigator Fellowship grant from the Novo Nordisk Foundation (NNF19OC0054973), a Consolidator Grant from the European Research Council (PATHORISC, ERC-2016-CoG 726417), a Research Infrastructure Grant from Carlsberg Fondet (CF20-0659), and an Instrument Grant from Brdr Hartmann Fonden (A35879), all awarded to PB, and by grants STA653/13 and STA653/14 from Deutsche Forschungsgemeinschaft to DS.

## Author contributions

**Marlene Reichel**: Conceptualization; Formal analysis; Validation; Investigation; Visualization; Methodology; Writing—review and editing. **Mathias Due Tankmar**: Formal analysis; Validation; Investigation; Visualization; Methodology; Writing—review and editing. **Sarah Rennie**: Conceptualization; Data curation; Software; Formal analysis; Investigation; Visualization; Methodology; Writing—review and editing. **Laura Arribas-Hernández**: Conceptualization; Supervision; Visualization; Writing—review and editing. **Martin Lewinski**: Formal analysis. **Tino Köster**: Supervision; Investigation. **Naiqi Wang**: Investigation; Writing—review and editing. **Anthony A Millar**: Conceptualization; Project administration; Writing—review and editing. **Dorothee Staiger**: Conceptualization; Supervision; Funding acquisition; Project administration; Writing—review and editing. **Peter Brodersen**: Conceptualization; Data curation; Supervision; Funding acquisition; Visualization; Methodology; Writing—original draft; Project administration; Writing—review and editing.

Source data underlying figure panels in this paper may have individual authorship assigned. Where available, figure panel/source data authorship is

listed in the following database record: biostudies:S-SCDT-10_1038-S44318-024-00312-0.

## Disclosure and competing interests statement

The authors declare no competing interests.

# Expanded View Figures

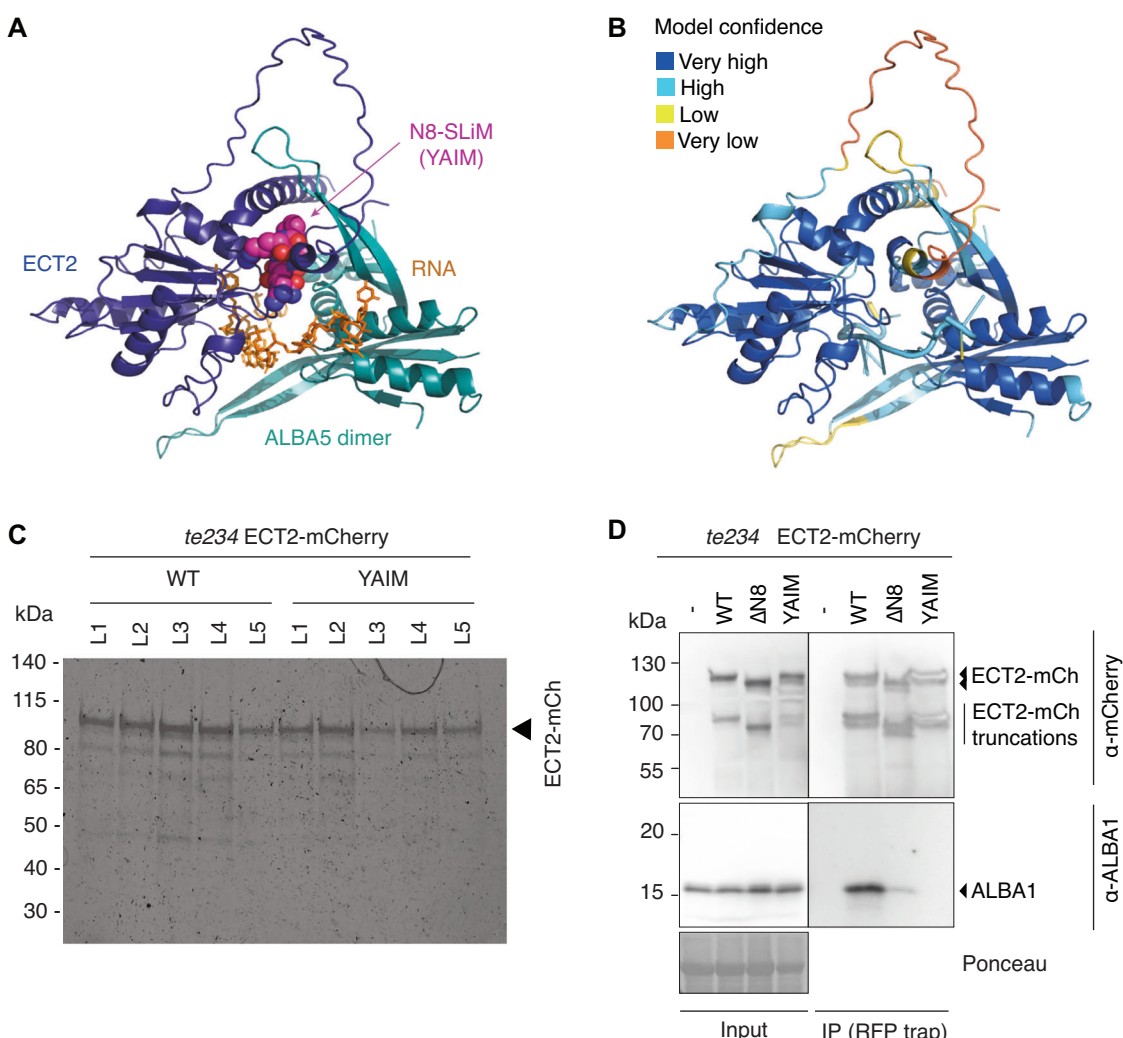

**Figure EV1. The YTH-ALBA Interacting Motif (YAIM) is central for ECT2-ALBA interaction (supporting data).**

(A) Alternative view of the AlphaFold3 model shown in Fig. 2B of the complex between ECT2 (YTH domain plus a YAIM-containing fragment of the N-terminal IDR), two ALBA5 subunits (ALBA domains only), and the 10-nt RNA [5′-AAA(m⁶A)CUUCUG-3′]. The YAIM is accentuated in space-fill mode (Magenta, C; Blue, N; Red, O), all other protein elements in cartoon mode, and the RNA in stick mode (supports Fig. 2B). (B) Same view of the model as in panel (A), but colored according to the predicted local distance difference test (pLDDT) score calculated by Alphafold3 to indicate model confidence on a local per-residue basis (Abramson et al, 2024) (supports Fig. 2C). (C) Silver staining of aliquots of immunopurified fractions used for LS-MS/MS analysis of differential enrichment in ECT2-mCherry and ECT2^YAIM-mCherry purifications (supports Fig. 2I). (D) Co-immunoprecipitation analysis of the ECT2-ALBA1 interaction. Nine-day-old seedlings from three independent transgenic lines expressing each ECT2-mCherry variant were pooled prior to mCherry immunoprecipitation and analysis by western blot. Source data are available online for this figure.

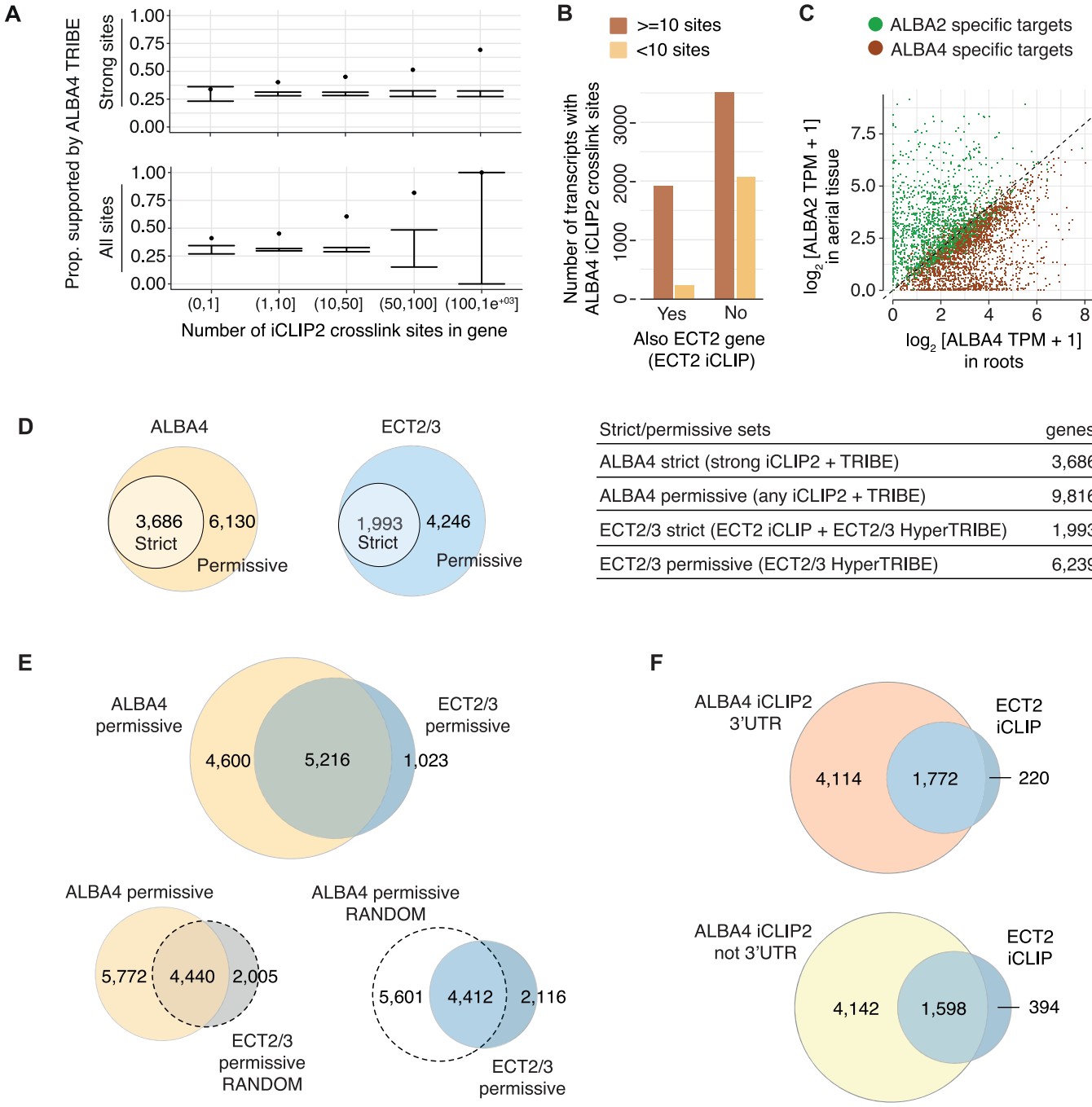

**Figure EV2. Analysis of ALBA2 and ALBA4 target sets (supporting data).**

(A) Proportion of genes supported by ALBA4-TRIBE according to the number of iCLIP2 crosslink sites in the gene. Points represent true proportions, and intervals represent background distribution based on sampling genes of similar expression levels to the target genes from ALBA4 iCLIP2. Error bars are based on 2.5 and 97.5% quantiles and $n = 177, 2282, 2855, 1299, 1132$ is the number of iCLIP2 crosslink sites considered in each group (from left to right). The top panel represents the strong set of replicated ALBA4 iCLIP2 peaks and bottom panel represents the full set of replicated ALBA4 iCLIP2 peaks. (B) Support of ALBA4 iCLIP2 crosslink site-containing genes according to whether the gene is also supported by ECT2 iCLIP crosslink site-containing genes. (C) Scatter plot comparing expression for targets specific to either ALBA4-TRIBE (roots) or ALBA2 HyperTRIBE (aerial tissues). Values represent the averaged $\log_2(\text{TPM} + 1)$ values across the Col-0 WT lines in each experiment. (D) Table showing defined strict and permissive gene sets for ECT2/3 and ALBA4. Venn diagrams provide visual representations of the ECT2/3 and ALBA4 strict and permissive gene sets. (E) Venn diagram for overlap between ALBA4 permissive and ECT2/3 permissive genes. Smaller Venn diagrams indicate overlap if either the ALBA4 or ECT2/3 permissive target sets were a randomly selected set of genes with a similar expression distribution to the true sets. (F) Venn diagrams for overlap between ECT2 iCLIP-derived target genes (strict) and ALBA4 iCLIP2 genes, where the ALBA4 set either contains or does not contain crosslink sites in its 3'-UTR.

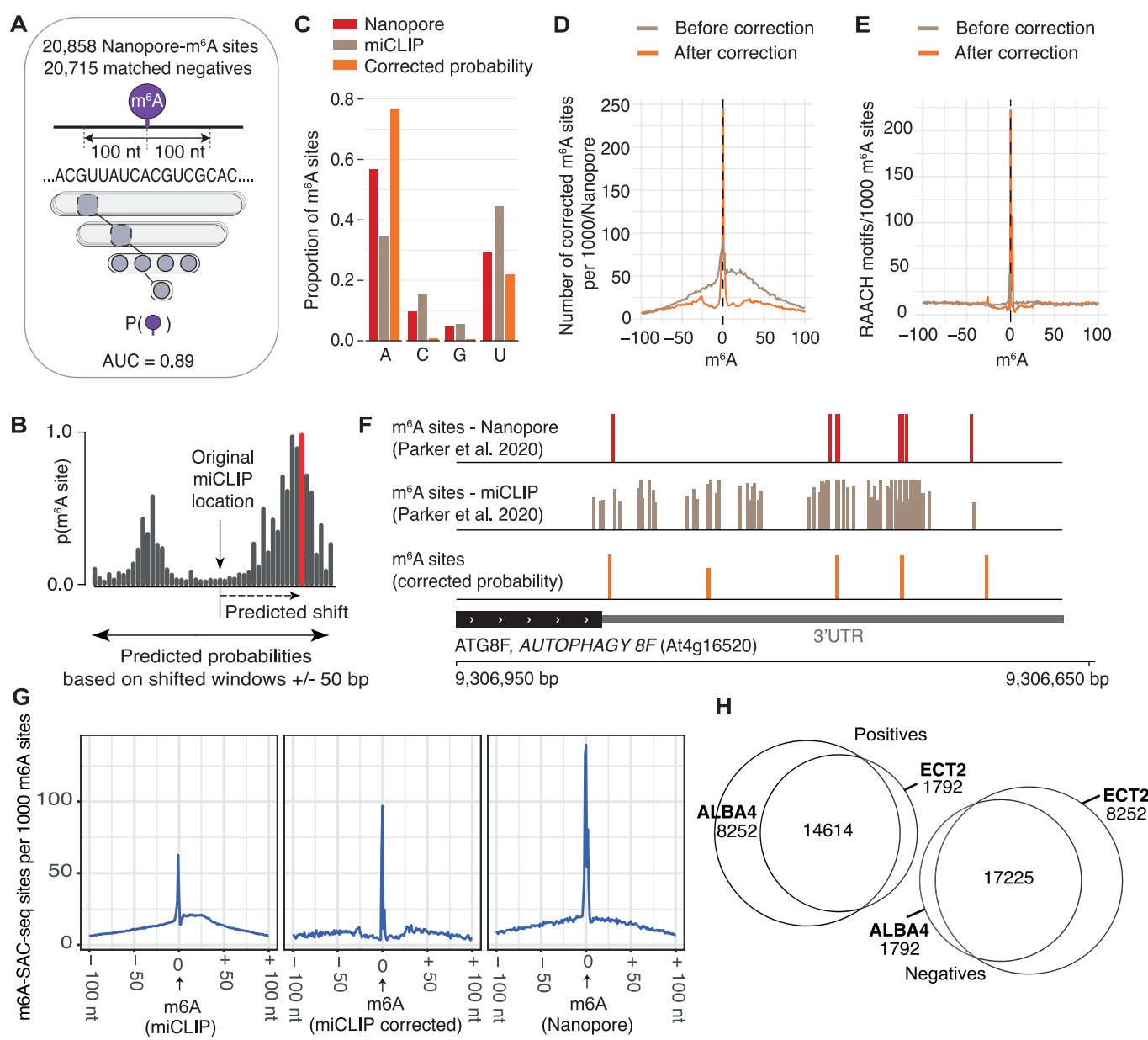

**Figure EV3.** A deep learning model to derive an augmented set of single-nucleotide-resolution m⁶A sites by integration of miCLIP and Nanopore data.

(A) Strategy for the m⁶A deep learning model. Each Nanopore m⁶A site was paired with a location-matched negative (see Methods for definition of location-matching), and sequences ±100 bp from the site were extracted and used as input to a convolutional neural network tasked with predicting the presence or absence of m⁶A in the center of the input sequence. AUC area under the curve. (B) Strategy for correcting the exact position of m⁶A sites based on the deep learning model. For each site and all individual positions within a ±50 bp window, a ±100 bp sequence was extracted and used as input to the m⁶A-network to predict the presence of m⁶A. The position with the strongest prediction was chosen as the corrected position for that site. (C) Reference base distribution of m⁶A sites as determined by Nanopore, miCLIP, or the corrected positions. (D) Enrichment of nanopore-determined m⁶A sites around the augmented set of m⁶A positions, before and after correction. (E) Enrichment of RRACH motifs per 1000 augmented m⁶A sites, before and after correction. (F) IGV view of a representative transcript, *AUTOPHAGY 8F* (At4g16520), showing the positions of m⁶A sites experimentally determined by either nanopore or miCLIP, and the positions resulting from the correction. (G) Enrichment of m⁶A sites identified by m⁶A-SAC-seq (Wang et al, 2024) around miCLIP sites before and after correction, and around nanopore-derived sites. (H) Annotation of m⁶A sites according to ECT2 and ALBA4 crosslink sites within 100 nt. Venn diagrams show overlap between the two proteins for both positives (bound) and negatives (non-bound) separately.

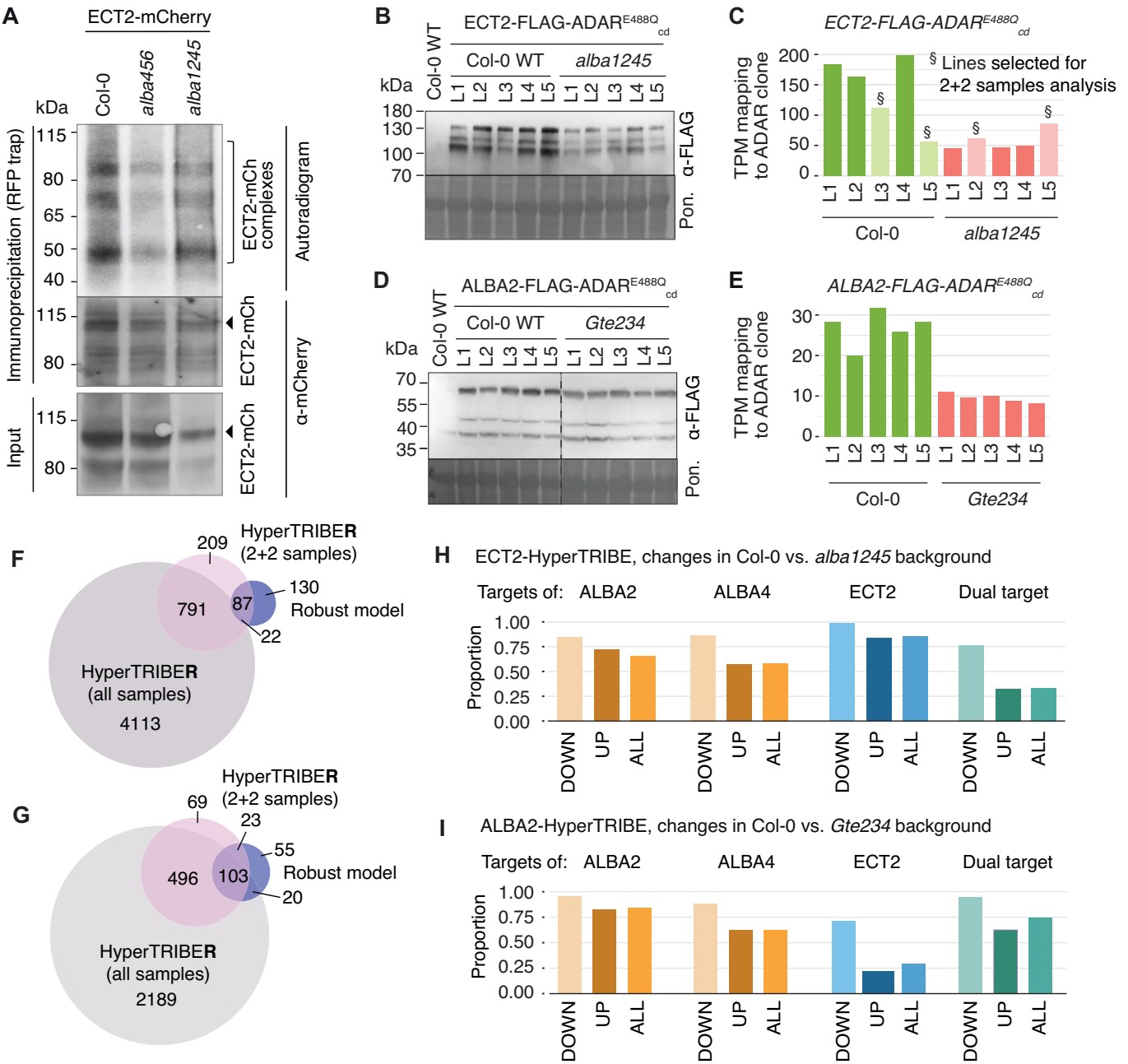

**Figure EV4. Mutual ECT-ALBA dependence for target mRNA binding.**

(A) Autoradiogram (top) of RNA-protein complexes from ECT2-mCherry expressed in Col-0, *alba1245*, or *alba456* after UV crosslinking and immunoprecipitation with RFP-trap beads. Marker positions and the location of the ECT2-mCherry-RNA adducts are indicated. Immunoblots against mCherry show the precipitated protein in the IP (middle) and input (bottom). Samples were pools of three independent lines for each genotype. (B) Western blot of the independent lines selected for HyperTRIBE analysis of ECT2 in Col-0 and *alba1245*. Ponceau staining was used as a loading control. (C) Detected number of transcripts per million of ECT2-FLAG-ADAR in lines used for HyperTRIBE. Light bars indicate lines selected for the two-sample analysis. (D) Western blot of the independent lines selected for HyperTRIBE analysis of ALBA2 in Col-0 and *gte234*. Ponceau staining was used as a loading control. (E) Detected number of transcripts per million of ALBA2-FLAG-ADAR in lines used for HyperTRIBE. (F) Overlap of individual sites detected as significant in ECT2-HT in Col-0 vs. *alba1245* according to three different approaches—HyperTRIBER pipeline (no correction), HyperTRIBER pipeline (using lines indicated in C) and developed a robust model. (G) As in (F) but counts depict containing genes for significant sites. (H–I) The proportion of sites significantly differentially edited by ECT2- (H) or ALBA2- (I) -FLAG-ADAR$^{E488Q}_{cd}$ fusions in Col-0 vs. the indicated mutant background (*alba1245* in H; *Gte234* in I), which are targets of ALBA2, ALBA4, ECT2, or both ALBA4 and ECT2 (dual-targets). UP/DOWN categories are defined according to whether the log$_2$ fold-change was negative (lower in the mutant background, DOWN) or positive (higher in the mutant background, UP). Source data are available online for this figure.

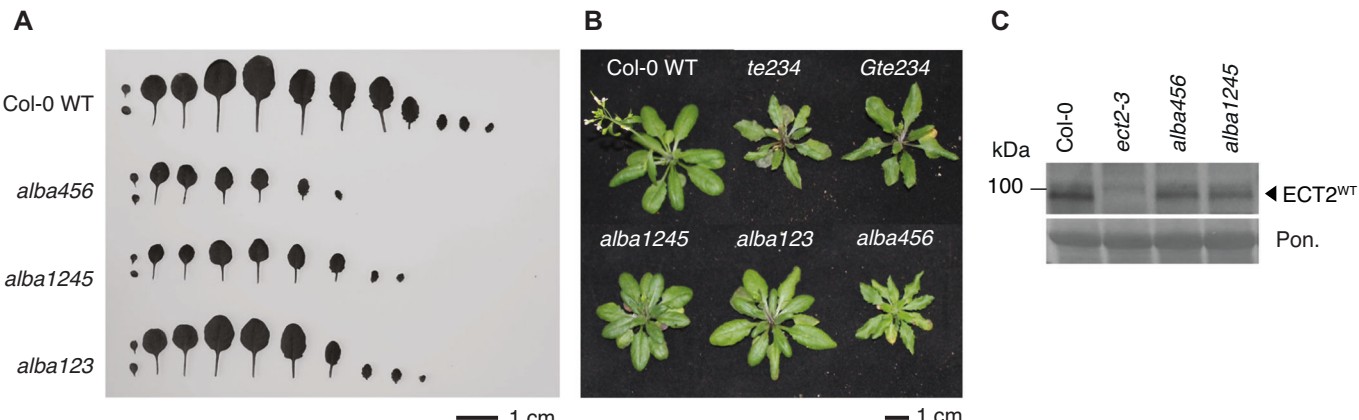

**Figure EV5.  Phenotypic analysis of *alba1245* and *alba456* mutants.**

(**A**) Leaf profiles of *alba* mutants at day 17 after germination. (**B**) Delayed flowering phenotype of higher-order *alba* and *ect* mutants (5-week-old, germinated directly on soil). (**C**) Protein blot of total lysates prepared from 10-day-old seedlings of the indicated genotypes, probed with ECT2-specific antisera (Arribas-Hernández et al, 2018). The arrow indicates the position of the ECT2^WT protein. Ponceau staining serves as the loading control. Source data are available online for this figure.

