## [Peer Review File · The EMBO Journal]

ALBA proteins facilitate cytoplasmic YTHDF-mediated reading of m6A in Arabidopsis

Peter Brodersen, Marlene Reichel, Mathias Tankmar, Sarah Rennie, Laura Arribas-Hernandez, Martin Lewinski, Tino Köster, Naiqi Wang, Anthony Millar, and Dorothee Staiger

Corresponding author(s): Peter Brodersen (pbrodersen@bio.ku.dk) , Dorothee Staiger (dorothee.staiger@uni-bielefeld.de), Sarah Rennie (sarah.rennie@bio.ku.dk), Anthony Millar (tony.millar@anu.edu.au)

Review Timeline:

Submission Date:	30th Jul 24
Editorial Decision:	2nd Sep 24
Revision Received:	16th Oct 24
Editorial Decision:	23rd Oct 24
Revision Received:	27th Oct 24
Accepted:	31st Oct 24

Editor: William Teale

Transaction Report:

Dear Peter,

Thank you again for the submission of your manuscript entitled "ALBA proteins facilitate cytoplasmic YTHDF-mediated reading of m6A in plants". We have now received the reports from the referees, which I copy below.

As you can see from their comments, all of them see your study as of interest and well conducted. There are, however, some points that will require your attention before your manuscript can be published in The EMBO Journal.

Based on the overall interest expressed in the reports, I would like to invite you to address the comments of all referees in a revised version of the manuscript. I should add that it is The EMBO Journal policy to allow only a single major round of revision and that it is therefore important to resolve the main concerns at this stage. I believe the concerns of the referees are reasonable and addressable, but please contact me if you have any questions, need further input on the referee comments or if you anticipate any problems in addressing any of their points. I am available for a Zoom call to discuss, for example, the feasibility of the in vitro experiment mentioned by Referee 1. Please, follow the instructions below when preparing your manuscript for resubmission.

I would also like to point out that as a matter of policy, competing manuscripts published during this period will not be taken into consideration in our assessment of the novelty presented by your study ("scooping" protection). We have extended this 'scooping protection policy' beyond the usual 3 month revision timeline to cover the period required for a full revision to address the essential experimental issues. Please contact me if you see a paper with related content published elsewhere to discuss the appropriate course of action.

Again, please contact me at any time during revision if you need any help or have further questions.

Thank you very much again for the opportunity to consider your work for publication. I look forward to your revision.

Best regards,

William

William Teale, Ph.D.
Editor
The EMBO Journal

When submitting your revised manuscript, please carefully review the instructions below and include the following items:

- 1) a .docx formatted version of the manuscript text (including legends for main figures, EV figures and tables). Please make sure that the changes are highlighted to be clearly visible.
- 2) individual production quality figure files as .eps, .tif, .jpg (one file per figure).
- 3) a .docx formatted letter INCLUDING the reviewers' reports and your detailed point-by-point response to their comments. As part of the EMBO Press transparent editorial process, the point-by-point response is part of the Review Process File (RPF), which will be published alongside your paper.
- 4) a complete author checklist, which you can download from our author guidelines ([https://wol-prod-cdn.literatumonline.com/pb-assets/embo-site/Author Checklist%20-%20EMBO%20J-1561436015657.xlsx](https://wol-prod-cdn.literatumonline.com/pb-assets/embo-site/Author%20Checklist%20-%20EMBO%20J-1561436015657.xlsx)). Please insert information in the checklist that is also reflected in the manuscript. The completed author checklist will also be part of the RPF.
- 5) Please note that all corresponding authors are required to supply an ORCID ID for their name upon submission of a revised manuscript.
- 6) We require a 'Data Availability' section after the Materials and Methods. Before submitting your revision, primary datasets produced in this study need to be deposited in an appropriate public database, and the accession numbers and database listed under 'Data Availability'. Please remember to provide a reviewer password if the datasets are not yet public (see <https://www.embopress.org/page/journal/14602075/authorguide#datadeposition>). If no data deposition in external databases is

needed for this paper, please then state in this section: This study includes no data deposited in external repositories. Note that the Data Availability Section is restricted to new primary data that are part of this study.

Note - All links should resolve to a page where the data can be accessed.

8) For data quantification: please specify the name of the statistical test used to generate error bars and P values, the number (n) of independent experiments (specify technical or biological replicates) underlying each data point and the test used to calculate p-values in each figure legend. The figure legends should contain a basic description of n, P and the test applied. Graphs must include a description of the bars and the error bars (s.d., s.e.m.).

9) We would also encourage you to include the source data for figure panels that show essential data. Numerical data can be provided as individual .xls or .csv files (including a tab describing the data). For 'blots' or microscopy, uncropped images should be submitted (using a zip archive or a single pdf per main figure if multiple images need to be supplied for one panel). Additional information on source data and instruction on how to label the files are available at .

10) We replaced Supplementary Information with Expanded View (EV) Figures and Tables that are collapsible/expandable online (see examples in <https://www.embopress.org/doi/10.15252/embj.201695874>). A maximum of 5 EV Figures can be typeset. EV Figures should be cited as 'Figure EV1, Figure EV2" etc. in the text and their respective legends should be included in the main text after the legends of regular figures.

12) Our journal encourages inclusion of *data citations in the reference list* to directly cite datasets that were re-used and obtained from public databases. Data citations in the article text are distinct from normal bibliographical citations and should directly link to the database records from which the data can be accessed. In the main text, data citations are formatted as follows: "Data ref: Smith et al, 2001" or "Data ref: NCBI Sequence Read Archive PRJNA342805, 2017". In the Reference list, data citations must be labeled with "[DATASET]". A data reference must provide the database name, accession number/identifiers and a resolvable link to the landing page from which the data can be accessed at the end of the reference. Further instructions are available at .

13) In order to increase the reproducibility and reach of your work, The EMBO Journal includes a table of reagents that were used in the study. Please provide this along with your revisions

Further instructions for preparing your revised manuscript:

We realize that it is difficult to revise to a specific deadline. In the interest of protecting the conceptual advance provided by the work, we recommend a revision within 3 months (1st Dec 2024). Please discuss the revision progress ahead of this time with the editor if you require more time to complete the revisions. Use the link below to submit your revision:

Referee #1:

The methylation of adenosines in mRNA is now well recognised to play important roles in post transcriptional regulation of multiple eukaryote developmental and environmental response programmes. The principle proteins that recognise the m6A modification do so via a "YTH domain", and such YTH domain proteins can be divided into YTHDC (largely nuclear) and YTHDF (cytoplasmic) families. Plants have an expanded family of the YTHDF class, but several in vitro assays have suggested that, compared to animal YTHDF proteins, the binding to m6A containing mRNAs is relatively weak - this has been a major source of discussion and confusion within the plant epitranscriptomics field. Focusing on the major Arabidopsis YTHDF protein, ECT2, this study resolves this by providing strong evidence that plant YTHDF proteins have a conserved interaction with ALBA proteins, and that the ALBA proteins bind to pyrimidine rich tracts found upstream of most m6A sites. Thus, it is the combination of both proteins acting together that is required for full binding to m6A containing transcripts. This raises the possibility that the binding of methylated transcripts is tunable via which of the ALBA protein family members are expressed in a tissue/condition, which YTHDF proteins are present, and the relative stoichiometry. Thus, a gradient of response outputs can potentially be generated to match the environmental/developmental conditions.

The data is clear and well presented - the interaction between ALBA family proteins and Arabidopsis YTHDF (ECT) proteins is demonstrated through co-IP, and the conserved motif within ECT2 (and land plant YTHDFs) responsible for the interaction is identified. The overlap between ECT2/3 and ALBA2/4 sites is also clear and convincing. A rather unique feature of plants with reduced m6A levels, or in which the m6A reader(s) have been knocked out, is the increased branching of trichomes. Thus, the fact that the knockout of the Arabidopsis ALBA proteins recapitulates this phenotype is particularly pleasing.

A very slight weakness is perhaps that in vitro binding of ALBA-ECT2 as a complex to m6A transcripts wasn't tested. However, reduced crosslinking to mRNAs in both alb456 and alba1245 backgrounds was demonstrated in vivo.

Overall, the manuscript is well written and the data is presented clearly. Importantly, the work provides an explanation for the rather poor m6A binding by plant YTHDFs on their own, and it opens many avenues of future research exploring how mRNA methylation may be used to regulate a plethora of developmental and environmental responses.

Referee #2:

The authors systematically analyzed the importance of the interactions between ECT2 and ALBAs in m6A binding and biological

function (seedling development in this case) in *Arabidopsis*. All genetic, biochemical, and molecular experiments were carried out well, and the data provided support their conclusions. This study is the first demonstration that the ALBA-YTHDF complex works together for m6A target recognition in plants, which proposes the novel concept of facilitated m6A reading by the ALBA-YTHDF module.

I have no critical concerns on this manuscript. One point that may be explained and/or discussed is how ALBA proteins were particularly selected for study. The authors described that ALBA proteins are highly enriched from IP-MS (Fig. 1F). There are many other spots co-purified with ECT2. What are those proteins? Are there any other potential interactors that might be involved in m6A binding? It would be informative if the list of other interactors is provided in Supplemental data.

In Fig. 2B and 2C, it would be more informative if the positions of amino acid residues mutated in ECT2YAIM are indicated by arrows.

Referee #3:

ALBA proteins facilitate cytoplasmic YTHDF-mediated reading of m6A in plants
by Reichel et al.

Reichel et al. discover a novel role of the ALBA RNA-binding proteins as co-factors of the m6A reader protein ECT2 in *Arabidopsis thaliana*. Starting out with a detailed characterisation of the intrinsically disordered region relevant for ECT2 function, the authors find interactions with both short and long ALBA proteins. They predict with AlphaFold3 modelling and then experimentally validate that a short linear motif in ECT2, termed YAIM, mediates the interaction between ECT2, ALBA and m6A mRNA. To determine the targets of this m6A-ECT-ALBA complex, the authors integrate multiple transcriptome-wide experiments, including iCLIP and TRIBE for ALBA2, ALBA4, and ECT2, revealing a large degree of overlap. iCLIP metaprofiles show that ECT2 binds directly at m6A sites whereas ALBA4 sits upstream, reflected in the distribution of putative binding motifs identified by different approaches. To demonstrate the mutual ALBA-ECT dependence in RNA binding, the authors perform reciprocal PNK assays and TRIBE experiments in mutant background of either protein, the latter analysed by targeted statistical approaches to correct for potential biases from expression differences. Finally, the authors demonstrate that inactivation of both ALBA and ECT proteins causes similar phenotypes.

Overall, the authors present a very interesting and relevant new mechanism for m6A reader activity in plants. They provide compelling evidence and support the major conclusions with orthogonal approaches, all within a framework that is carefully designed and well-controlled. The study builds on an elegant combination of experimental and computational approaches, including multiple transcriptome-wide assays and machine learning on the obtained data.

Comments:

Several analyses that compare the binding between ECT and ALBA proteins (e.g., in Figure 5) rely on a matched background set. I could not find how this was generated and what it corrects for. In the quantitative comparisons, it is important to control for expression biases. For instance, the authors state that "the enrichment of ALBA4 peaks at m6A-sites was much more pronounced when considering peaks in ECT2 targets compared to nontargets." A confounding factor in this analysis could be that ECT2 binding is more likely to be detected on more abundant transcripts, hence achieving also higher ALBA4 signal. Did the authors control for such biases?

The positional profiles of ECT2, ALBA4 and m6A indicate that ALBA4 binds upstream of ECT2 which in turn binds directly on the m6A sites. This is reflected in the motif distribution in Figure 5G in which the U/Y-rich motifs accumulate upstream of ECT2 binding sites. However, the motifs appear on both sides of m6A sites. This bimodal pattern is also visible in Figure 6E for the ECT2+ALBA4-preferred motifs. Can the authors explain this?

In the methods sections, the authors describe a neural network used to curate a compendium of m6A sites for *Arabidopsis thaliana*. This is a very valuable resource for other researchers and should be made available. How were the negative sites chosen for this analysis? The methods description mentions the wrong supplementary figure (should be Figure S10).

I encourage the authors to deposit the generated data in a database like NCBI GEO in which it can be accompanied with processed data. In my experience, this strongly increases its usability for other researchers.

Figure 6A: What is the overlap of ECT2+ and ALBA4+ m6A sites used for training the neural network?

Other:

Figure 5B lacks scale bars to give a feeling about the distance between the m6A sites and the ECT2 and ALBA4 binding sites.

Figure 5E: I think the Y-axis label should be "per ECT2 crosslink sites" (not per m6A sites).

RESPONSE TO REVIEWERS

We thank all three reviewers for their thorough and overall positive evaluation of our manuscript. Below, we elaborate on the changes we have made to improve the manuscript in response to the points raised by the reviewers. For clarity, the reviewers' comments are in black and our responses are in blue.

Referee #1:

The methylation of adenosines in mRNA is now well recognised to play important roles in post transcriptional regulation of multiple eukaryote developmental and environmental response programmes. The principle proteins that recognise the m⁶A modification do so via a "YTH domain", and such YTH domain proteins can be divided into YTHDC (largely nuclear) and YTHDF (cytoplasmic) families. Plants have an expanded family of the YTHDF class, but several *in vitro* assays have suggested that, compared to animal YTHDF proteins, the binding to m⁶A containing mRNAs is relatively weak - this has been a major source of discussion and confusion within the plant epitranscriptomics field. Focusing on the major Arabidopsis YTHDF protein, ECT2, this study resolves this by providing strong evidence that plant YTHDF proteins have a conserved interaction with ALBA proteins, and that the ALBA proteins bind to pyrimidine rich tracts found upstream of most m⁶A sites. Thus, it is the combination of both proteins acting together that is required for full binding to m⁶A containing transcripts. This raises the possibility that the binding of methylated transcripts is tunable via which of the ALBA protein family members are expressed in a tissue/condition, which YTHDF proteins are present, and the relative stoichiometry. Thus, a gradient of response outputs can potentially be generated to match the environmental/developmental conditions.

The data is clear and well presented - the interaction between ALBA family proteins and Arabidopsis YTHDF (ECT) proteins is demonstrated through co-IP, and the conserved motif within ECT2 (and land plant YTHDFs) responsible for the interaction is identified. The overlap between ECT2/3 and ALBA2/4 sites is also clear and convincing. A rather unique feature of plants with reduced m⁶A levels, or in which the m⁶A reader(s) have been knocked out, is the increased branching of trichomes. Thus, the fact that the knockout of the Arabidopsis ALBA proteins recapitulates this phenotype is particularly pleasing. A very slight weakness is perhaps that *in vitro* binding of ALBA-ECT2 as a complex to m⁶A transcripts wasn't tested. However, reduced crosslinking to mRNAs in both *alba456* and *alba1245* backgrounds was demonstrated *in vivo*.

Overall, the manuscript is well written and the data is presented clearly. Importantly, the work provides an explanation for the rather poor m⁶A binding by plant YTHDFs on their own, and it opens many avenues of future research exploring how mRNA methylation may be used to regulate a plethora of developmental and environmental responses.

We thank referee 1 for the thorough assessment of our work.

While we see the relevance of the *in vitro* binding assay, we also believe that the *in vivo* crosslinking and ECT2 HyperTRIBE experiments in wild type, *alba1245* and *alba456* mutant backgrounds provide clear evidence for the requirement for ALBA proteins for full m⁶A-binding activity of ECT2 *in vivo*; the main conclusion of the paper.

Referee #2:

The authors systematically analyzed the importance of the interactions between ECT2 and ALBAs in m⁶A binding and biological function (seedling development in this case) in Arabidopsis. All genetic, biochemical, and molecular experiments were carried out well, and the data provided support their conclusions. This study is the first demonstration that the ALBA-YTHDF complex works together for m⁶A target recognition in plants, which proposes the novel concept of facilitated m⁶A reading by the ALBA-YTHDF module.

I have no critical concerns on this manuscript. One point that may be explained and/or discussed is how ALBA proteins were particularly selected for study. The authors described that ALBA proteins are highly enriched from IP-MS (Fig. 1F). There are many other spots co-purified with ECT2. What are those proteins? Are there any other potential interactors that might be involved in m⁶A binding? It would be informative if the list of other interactors is provided in Supplemental data. In Fig. 2B and 2C, it would be more informative if the positions of amino acid residues mutated in ECT2YAIM are indicated by arrows.

We thank referee 2 for the positive assessment of our study.

We have now clarified that the ALBA proteins were chosen for further study because the combination of the following points apply to them:

- (1) They are highly abundant in all ECT2 and ECT3 IP-MS experiments, almost as abundant as the ECTs themselves, suggesting robust interaction.**
- (2) Their abundance is clearly reduced in the ECT2^{ΔN8} mutant.**
- (3) ALBA proteins are known to have RNA-binding activity; an appealing property given the reduced RNA association of the ECT2^{ΔN8} mutant *in vivo* (Fig. 1E).**

These points were sufficient to catch our interest for follow-up study.

It is true that other proteins satisfy points (2) and (3), but the ALBAs stand out with their abundance relative to the purified ECT protein. Nonetheless, the reviewer is right that we cannot exclude participation of other RNA-binding proteins in m⁶A recognition; we can only say for sure that the ALBA proteins have such a function. We have now added a sentence to this effect at the very end of the Discussion.

Identities of co-purified proteins and their peptide abundances are given in Dataset EV1. Some of the purifications (HA-ECT2 vs Col-0; ECT3-Venus vs Col-0, ECT1-TFP vs Col-0) have been published in previous reports (Tankmar et al. (2023), EMBO Reports 24(12): 57741. doi: 10.15252/embr.202357741; Flores-Télez et al. (2023), PLOS Genetics 19 (10): e1010980. doi: 10.1371/journal.pgen.1010980), and identities of the co-purified proteins are given in the supplemental material of those reports.

Regarding the AlphaFold3 model shown in Fig. 2B/C, we have highlighted the entire YAIM in pink, and since the Alanine mutations touch most residues within the YAIM, we think the colour code does a good job of showing where the mutations are. It becomes a bit confusing to look at with the many arrows pointing to positions all along the pink segment.

Referee #3:

ALBA proteins facilitate cytoplasmic YTHDF-mediated reading of m⁶A in plants by Reichel et al.

Reichel et al. discover a novel role of the ALBA RNA-binding proteins as co-factors of the m⁶A reader protein ECT2 in *Arabidopsis thaliana*. Starting out with a detailed characterisation of the intrinsically disordered region relevant for ECT2 function, the authors find interactions with both short and long ALBA proteins. They predict with AlphaFold3 modelling and then experimentally validate that a short linear motif in ECT2, termed YAIM, mediates the interaction between ECT2, ALBA and m⁶A mRNA. To determine the targets of this m⁶A-ECT-ALBA complex, the authors integrate multiple transcriptome-wide experiments, including iCLIP and TRIBE for ALBA2, ALBA4, and ECT2, revealing a large degree of overlap. iCLIP metaprofiles show that ECT2 binds directly at m⁶A sites whereas ALBA4 sits upstream, reflected in the distribution of putative binding motifs identified by different approaches. To demonstrate the mutual ALBA-ECT dependence in RNA binding, the authors perform reciprocal PNK assays and TRIBE experiments in mutant background of either protein, the latter analysed by targeted statistical approaches to correct for potential biases from expression differences. Finally, the authors demonstrate that inactivation of both ALBA and ECT proteins causes similar phenotypes.

Overall, the authors present a very interesting and relevant new mechanism for m⁶A reader activity in plants. They provide compelling evidence and support the major conclusions with orthogonal approaches, all within a framework that is carefully designed and well-controlled. The study builds on an elegant combination of experimental and computational approaches, including multiple transcriptome-wide assays and machine learning on the obtained data.

We appreciate the reviewer's kind words and thoughtful feedback on our work. Please see responses to individual comments below.

Comments:

Several analyses that compare the binding between ECT and ALBA proteins (e.g., in Figure 5) rely on a matched background set. I could not find how this was generated and what it corrects for.

We apologize for the lack of clarity regarding the matched backgrounds. The matched background refers to the distribution of the locations in the background set, which should be as close to the set of interest as possible (for example, ECT2 binding sites in the 3'UTR near the stop codon should be matched with similarly located sites on a random gene with a similar expression level). The strategy is described in our previous publication (Arribas-Hernández eLife 10 (2021):e72375. doi: 10.7554/eLife.72375), but we have now added methods text to clarify:

“Definition of matched backgrounds sets

For sites of interest (iCLIP crosslink sites, m⁶A site positions, etc.) enrichment of surrounding features (motifs, sites of interest) was assessed by comparing

to the same features around matched background sets. These sets were calculated similarly to our previous publication (Arribas-Hernández eLife 10 (2021):e72375. doi: 10.7554/eLife.72375), where, briefly, each site of interest was paired with a site of similar proportion along the relevant transcript feature (5'-UTR, CDS or 3'-UTR). To avoid expression bias, the pool of background genes was limited to those of the set of interest, or alternatively based on a set of genes selected to have a similar expression distribution.”

In the quantitative comparisons, it is important to control for expression biases. For instance, the authors state that “the enrichment of ALBA4 peaks at m6A-sites was much more pronounced when considering peaks in ECT2 targets compared to nontargets.” A confounding factor in this analysis could be that ECT2 binding is more likely to be detected on more abundant transcripts, hence achieving also higher ALBA4 signal. Did the authors control for such biases?

We thank the reviewer for the highly relevant comment. We have made efforts to account for abundance in the analyses in the paper. With respect to the specific analysis in referred to Figure 5F, the “background” sites were selected randomly from genes which are in the ECT2 permissive set. This set should therefore be matched in expression to the ECT2 target set plotted. Similar strategies were also used for the other figures – for example, for Figure 5E the background sites only fall on genes which are strict ECT2 targets (and again with a matched location distribution on the transcript).

We hope the reviewer agrees that this strategy is robust and convincing. For proposed changes to the Methods section, please see the text from the previous comment.

The positional profiles of ECT2, ALBA4 and m6A indicate that ALBA4 binds upstream of ECT2 which in turn binds directly on the m6A sites. This is reflected in the motif distribution in Figure 5G in which the U/Y-rich motifs accumulate upstream of ECT2 binding sites. However, the motifs appear on both sides of m6A sites. This bimodal pattern is also visible in Figure 6E for the ECT2+ALBA4-preferred motifs. Can the authors explain this?

We think that the symmetrical appearance of U/Y-rich elements around m⁶A may be explained by the fact that these sequence elements are not only important for m⁶A reading via ALBA proteins, but also for m⁶A writing by the nuclear N⁶-adenosine methyl transferase. Indeed, in one of our previous studies (Arribas-Hernández eLife 10 (2021):e72375. doi: 10.7554/eLife.72375), we found that U/Y-rich elements in the regions of 10-50 nt upstream AND downstream of m⁶A were strong predictors of whether a certain site is a *bona fide* m6A site or not. The biochemical explanation for this observation is likely to involve U/Y-binding proteins as subunits of the methyltransferase: in metazoan cells, the uridine-binding protein RBM15/Spenito is part of the methyltransferase and is required for m⁶A deposition. In plants, an RBM15 homologue (FPA) co-purifies with core methyltransferase subunits, but genetic inactivation of FPA does not lead to detectable decreases in m⁶A in mRNA, perhaps suggesting redundancy with other uridine-binding proteins.

In the methods sections, the authors describe a neural network used to curate a compendium of m⁶A sites for *Arabidopsis thaliana*. This is a very valuable resource for other researchers and should be made available.

The neural network data, including all models, training and testing data sets (in FASTA format), coordinates of m⁶A sites and predictions are all available on Zenodo (10.5281/zenodo.11241987). In addition, all m⁶A sites, annotated according to their predicted probabilities (where applicable) and the probability of binding bound by ECT2 or ALBA are provided as Dataset EV5.

How were the negative sites chosen for this analysis?

Thank you for your question. The set of ~48K m⁶A sites were annotated according to whether they are bound by ECT2 or ALBA4 (defined as a robust crosslink site within 100 nt of the m⁶A site). Therefore, for each task (Task 1: ECT2 presence or absence, Task 2: ALBA4 presence or absence), the negatives are defined as m⁶A sites which were not bound by the relevant protein.

We have clarified the method section accordingly:

“Convolutional neural network based de-novo motif detection

We annotated each of 41,883 m⁶A sites as bound or unbound according to overlap of either ECT2 iCLIP or ALBA iCLIP sites within 100 nt. For each task (task 1: predicting the presence of ECT2 binding, task 2: predicting the presence of ALBA4 binding), we defined the positives as m⁶A sites bound by the relevant protein and the negatives as unbound m⁶A sites.

The methods description mentions the wrong supplementary figure (should be Figure S10).

We thank the reviewer for this observation - the figure reference has been updated accordingly.

I encourage the authors to deposit the generated data in a database like NCBI GEO in which it can be accompanied with processed data. In my experience, this strongly increases its usability for other researchers.

We thank the reviewer for their comment and agree that supplying both the raw and processed data is important. In addition to supplementary files containing processed datasets, we have created a repository on Zenodo (10.5281/zenodo.11241987) containing all processed data objects (as R objects) - these R objects can be used as starting points for the supplied scripts in the Github repository (https://github.com/sarah-ku/ALBA_YTH_arabidopsis) for reproducing the analyses from our paper. In addition, the raw reads for all sequencing data have been deposited to ENA (European Nucleotide Archive).

Figure 6A: What is the overlap of ECT2+ and ALBA4+ m⁶A sites used for training the neural network?

We have included a Venn diagram in the supplementary showing the overlap between m⁶A sites defined as bound by ECT2 and those bound by ALBA4, separately for both the positive and negative sets. This is now Figure EV3H. The main text also includes the following:

“Although there was a large overlap between the two proteins, there was a sizable set of bound sites unique to each protein (Figure EV3).”

Other:

Figure 5B lacks scale bars to give a feeling about the distance between the m⁶A sites and the ECT2 and ALBA4 binding sites.

Figure 5E: I think the Y-axis label should be "per ECT2 crosslink sites" (not per m⁶A sites).

We thank the reviewer for their observations. We have added a scale bar to Figure 5B and have corrected the y-axis label in Figure 5E.

Dear Peter,

Thank you for submitting the revised version of your manuscript, which addresses the concerns of the referees. This revised version has now been re-reviewed; I attach the second referee reports to the bottom of this mail. As you will see, you have addressed the referees' concerns to their satisfaction. Before I can finally accept the manuscript, there are some remaining editorial points which need to be addressed. In this regard, would you please:

- limit the number of keywords to five,
- remove the author credit section from the manuscript file,
- change nomenclature of tables to Dataset EVx throughout each Excel file (instead of legends labeled as Table S1, S2...); Table EV1 should be renamed as Dataset EV6,
- add the title of the manuscript to the title page (Appendix for ALBA proteins facilitate cytoplasmic YTHDF-mediated reading of m6A in plants),
- provide specific URLs for PRJEB71752 and PXD052232 datasets in the data availability statement,
- provide exact p values in the legends of figures 1c; 2f; 4d, g-j; 7d-e, g,
- correct mismatch in figure 2f between the annotated p values in the figure legend and the annotated p values in the figure file,
- define box plots need in terms of minima, maxima, centre, bounds of box and whiskers, and percentile in the legends of figures 6b; 7d-e,
- define 'n' in the legends of figures 1f-g; 2i; 6b; 7d-e; EV 2a,
- define error bars in the legend of figure EV 2a, and
- rename the "MATERIALS AND METHODS" section as the "METHODS" section.

I look forward to receiving these changes. EMBO Press is an editorially independent publishing platform for the development of EMBO scientific publications.

Best wishes,

William

William Teale, PhD
Editor
The EMBO Journal
w.teale@embojournal.org

Please remember: Digital image enhancement is acceptable practice, as long as it accurately represents the original data and conforms to community standards. If a figure has been subjected to significant electronic manipulation, this must be noted in the figure legend or in the 'Materials and Methods' section. The editors reserve the right to request original versions of figures and

the original images that were used to assemble the figure.

We realize that it is difficult to revise to a specific deadline. In the interest of protecting the conceptual advance provided by the work, we recommend a revision within 3 months (21st Jan 2025). Please discuss the revision progress ahead of this time with the editor if you require more time to complete the revisions. Use the link below to submit your revision:

Referee #1:

I have read the revised manuscript and am happy that all comments have been addressed. This is an important piece of work that will influence the direction of plant m6A research for many years to come.

Referee #2:

The authors addressed all of my comments and suggestions, which further support and strengthen the conclusion. I have no further concerns.

Referee #3:

The authors thoroughly addressed all concerns raised.

All editorial and formatting issues were resolved by the authors.

Dear Peter,

I am pleased to inform you that your manuscript has been accepted for publication in the EMBO Journal.

Congratulations to you and your team for a really exciting collection of experiments!

Best wishes,

William

William Teale, PhD
Editor
The EMBO Journal
w.teale@embojournal.org
